# Lipoate-binding proteins and specific lipoate-protein ligases in microbial sulfur oxidation reveal an atpyical role for an old cofactor

Xinyun Cao[1†‡], Tobias Koch[2†], Lydia Steffens[2], Julia Finkensieper[2], Renate Zigann[2], John E Cronan[1,3], Christiane Dahl[2*]

[1]Department of Biochemistry, University of Illinois, Urbana, United States; [2]Institut für Mikrobiologie and Biotechnologie, Rheinische Friedrich-Wilhelms-Universität Bonn, Bonn, Germany; [3]Department of Microbiology, University of Illinois, Urbana, United States

**\*For correspondence:**
ChDahl@uni-bonn.de

[†]These authors contributed equally to this work

**Present address:** [‡]Department of Biochemistry, University of Wisconsin-Madison, Madison, United States

**Abstract** Many Bacteria and Archaea employ the heterodisulfide reductase (Hdr)-like sulfur oxidation pathway. The relevant genes are inevitably associated with genes encoding lipoate-binding proteins (LbpA). Here, deletion of the gene identified LbpA as an essential component of the Hdr-like sulfur-oxidizing system in the Alphaproteobacterium *Hyphomicrobium denitrificans*. Thus, a biological function was established for the universally conserved cofactor lipoate that is markedly different from its canonical roles in central metabolism. LbpAs likely function as sulfur-binding entities presenting substrate to different catalytic sites of the Hdr-like complex, similar to the substrate-channeling function of lipoate in carbon-metabolizing multienzyme complexes, for example pyruvate dehydrogenase. LbpAs serve a specific function in sulfur oxidation, cannot functionally replace the related GcvH protein in *Bacillus subtilis* and are not modified by the canonical *E. coli* and *B. subtilis* lipoyl attachment machineries. Instead, LplA-like lipoate-protein ligases encoded in or in immediate vicinity of *hdr-lpbA* gene clusters act specifically on these proteins.
DOI: https://doi.org/10.7554/eLife.37439.001

## Introduction

Lipoic acid (1,2-dithiolane-3-pentanoic acid or 6,8-thioactic acid, *Figure 1*) is a highly conserved organosulfur cofactor found in all three domains of life. So far, only five lipoate-dependent multienzyme complexes have been characterized: three α-ketoacid dehydrogenases (pyruvate dehydrogenase, 2-oxoglutarate dehydrogenase, branched-chain 2-oxoacid dehydrogenase), acetoin dehydrogenase and the glycine cleavage complex (*Cronan, 2016*; *Cronan et al., 2005*). In all of these enzymes, a specific subunit is modified by lipoic acid attachment to the ε-amino groups of specific lysine residues within conserved lipoyl domains. In the five characterized lipoylated enzyme complexes, lipoate acts both as an electrophile that binds to reaction intermediates (via a thioester or thioether bond) and as a swinging arm that channels the bound substrate between the active sites of different subunits. During catalysis, the intramolecular disulfide bond of lipoate cycles between oxidized lipoamide and reduced dihydrolipoamide (*Figure 1*) (*Reed and Hackert, 1990*).

Comparative genome mining of sulfur-oxidizing prokaryotes (*Dahl, 2015*; *Quatrini et al., 2009*; *Venceslau et al., 2014*), transcriptomic and proteomic studies (*Latorre et al., 2016*; *Mangold et al., 2011*; *Osorio et al., 2013*; *Ouyang et al., 2013*; *Quatrini et al., 2009*) as well as first biochemical evidence (*Boughanemi et al., 2016*) indicated that a large group of archaeal and

**Figure 1.** Structures of lipoic acid and its reduced derivative dihydrolipoic acid.
DOI: https://doi.org/10.7554/eLife.37439.002

bacterial sulfur oxidizers utilize a novel pathway of sulfur oxidation involving a heterodisulfide reductase (Hdr)-like enzyme complex resembling the HdrABC (*Wagner et al., 2017*) of methanogens, although a coenzyme M-coenzyme B (CoM-S-S-CoB) heterodisulfide is not present in the sulfur oxidizers. Recently, we provided conclusive genetic evidence that this complex is indeed essential for oxidation of thiosulfate to sulfate in *Hyphomicrobium denitrificans* (*Koch and Dahl, 2018*). In this Alphaproteobacterium, thiosulfate is an intermediate of dimethylsulfide (DMS) degradation and a Δ*hdr* strain was also unable to grow on the volatile organosulfur compound (*Koch and Dahl, 2018*). The sulfur-oxidizing prokaryote genes (*hdrC1B1A-hyp-hdrC2B2*) encoding the Hdr-like complex are associated with genes encoding novel lipoate-binding proteins resembling GvcH, the lipoate-binding component of the bacterial glycine cleavage system (*Ehrenfeld et al., 2013*; *Liu et al., 2014*). The first analyses of these gene clusters furthermore indicated the presence of genes potentially encoding enzymes responsible for the biosynthesis of lipoamide-containing proteins (*Liu et al., 2014*).

In general, two different mechanisms of posttranslational modification of proteins with lipoate can be distinguished: de novo lipoate biosynthesis and lipoate scavenging (*Spalding and Prigge, 2010*). In *E. coli*, using exogenous lipoate requires LplA, a two domain lipoate-protein ligase. Lipoate is first activated to lipoyl-AMP at the expense of ATP. The N-terminal domain then transfers the lipoyl-group to the target protein. In contrast, endogenous lipoate synthesis proceeds through two steps: i. LipB-catalyzed transfer of an octanoyl group from octanoyl acyl carrier protein (octanoyl-ACP) to the apoprotein and ii. insertion of two sulfur atoms at $C_6$ and $C_8$ of the octanoyl chain catalyzed by the radical SAM protein lipoyl synthase (LipA).

Here, we set out to test for functional linkage between *hdr*-like sulfur oxidation genes with genes for lipoate-binding proteins together with the enzymes putatively involved in their modification. The functional role of lipoate-binding proteins in sulfur compound oxidation was assessed in the genetically accessible model organism *H. denitrificans*. Phylogenetic analyses identified two clearly distinguishable subgroups of the putative lipoate-binding proteins. Proteins from both groups could only be modified by LplA-like proteins encoded in immediate vicinity with other *hdr*-like genes. Moreover, the LplA-like protein has single domain architecture whereas other lipoate ligases are two domain proteins.

## Results

### Hdr, lipoate-binding protein (LbpA) and LbpA maturation genes

Database searches revealed 56 different cultivated genome-sequenced prokaryotes from one archaeal and six bacterial phyla as containing typical *hdr*-like gene loci (*Figure 2—source data 1*). With very few exceptions all these organisms are either established chemo- and photolithotrophs oxidizing reduced inorganic sulfur compounds or they have been reported as oxidizing mineral sulfides (FeS, $FeS_2$ and others) or volatile organic sulfur compounds (e.g. dimethylsulfide). The remaining species were isolated from sulfur-dominated habitats such as solfataras or ocean sediments and therefore are likely to be sulfur compound oxidizers (*Figure 2—source data 1*).

In all cases, the cluster of *hdr*-like genes is associated with at least one, in most cases with two and in a few cases with three genes for putative lipoate-binding proteins resembling classical glycine cleavage system H (GcvH) proteins (*Figure 2*, *Figure 2—source data 1*). We name this single lipoyl

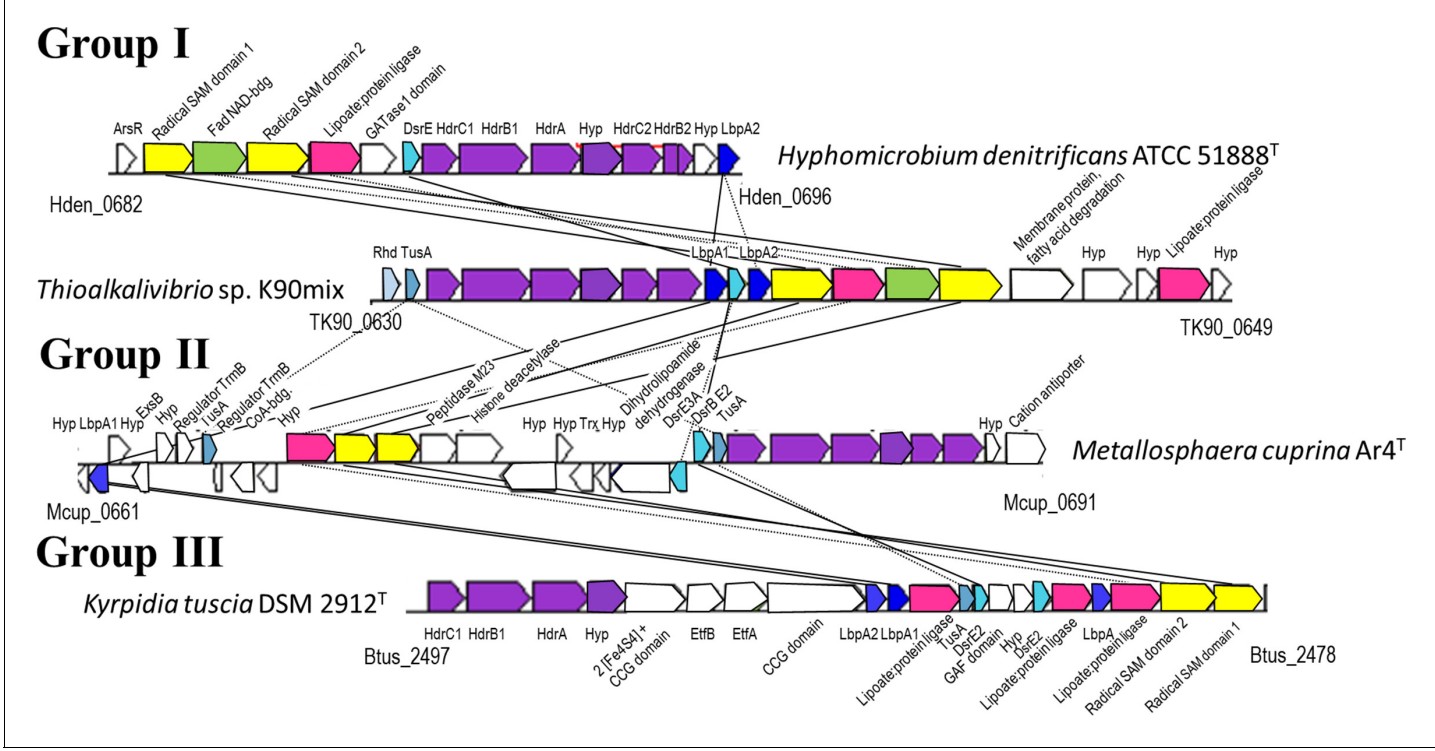

**Figure 2.** Representative operon structures/gene clusters from organisms containing *hdr*-like and *lbpA* genes. Homologous genes are colored the same between organisms. Genes in white are not conserved between the groups. The Kegg/NCBI locus tag identifiers for the first and last genes are shown below each cluster. GATase1, glutamine amidotransferase class I; DsrE, -E2, -E3A, DsrE-like sulfurtransferases (**Dahl, 2015**); Rhd, rhodanese-like sulfurtransferase (**Bordo and Bork, 2002**); Hyp, hypothetical protein; TusA, TusA-like sulfur carrier protein (**Dahl, 2015**; **Dahl et al., 2013**); CCG domain, domain with CCG signature motif of non-cubane [4Fe-4S] cluster-binding heterodisulfide reductase active sites (**Wagner et al., 2017**); Etf, electron-transferring flavoprotein (**Garcia Costas et al., 2017**).

DOI: https://doi.org/10.7554/eLife.37439.003

The following source data and figure supplement are available for figure 2:

**Source data 1.** Occurrence of *hdr*-like, *lbpA* and *lbpA* maturation genes in genome-sequenced prokaryotes.
DOI: https://doi.org/10.7554/eLife.37439.005

**Figure supplement 1.** Sequence alignment of Hdr-associated lipoate-binding proteins from seven archaeal and bacterial phyla.
DOI: https://doi.org/10.7554/eLife.37439.004

domain protein LbpA (lipoate binding protein A). In almost all cases, the *hdr-lbpA* arrangement is linked in the same gene cluster structure with genes encoding proteins probably involved in the biosynthesis of lipoate binding proteins. These include potential lipoate-protein ligases (LplA) and two different radical SAM domain containing proteins (RadSAM1 and RadSAM2). If not located in the same operon or in immediate vicinity of the *hdr-lbpA* genes, the biosynthetic enzymes are encoded elsewhere in the genome (*Figure 2—source data 1*). Only three of the bacterial genomes lack homologs of the potential biosynthetic genes. The close genomic association of lipoate-binding proteins with maturation enzymes together with the observation that virtually all organisms analyzed contain enzymes catalyzing the canonical pathways for biosynthesis of lipoylated proteins (*Figure 2—source data 1*) indicate the possibility of a specific biosynthetic pathway required for maturation of Hdr-associated lipoate-binding proteins.

Three different types of arrangements of *hdr*-like, *lbpA* and biosynthetic genes can be differentiated (*Figure 2*). In the most common group (Group I) found in all analyzed Proteobacteria and Aquificae, the maturation genes are strictly located in the same operon as the *hdr*-like and *lbpA* genes. The Group II arrangement is typical for the archaeal family Sulfolobaceae. The genes are scattered about a wider region and located in several different operons. Group III presents yet a different arrangement and is found in Actinobacteria and Firmicutes where genes encoding Hdr-like and two or three lipoate-binding proteins are linked with genes for electron-transferring flavoproteins (EtfBA)

(*Garcia Costas et al., 2017*). Although a close functional association between the encoded proteins is strongly indicated for all three groups, differences in the interplay of the encoded proteins are certainly to be expected.

## Sequence alignments and phylogenetic analyses for Hdr-associated lipoate-binding proteins

An alignment was built for 95 LbpA proteins encoded in or in immediate vicinity of *hdr*-like gene clusters. All these proteins share a strictly conserved lysine residue (*Figure 2—figure supplement 1*) known to be required for lipoate attachment (*Spalding and Prigge, 2010*). With the exception of some archaeal LbpAs, the most conspicuous common features of the new LbpA proteins are two strictly conserved cysteine residues that GcvH proteins lack, one cysteine is located near the N-terminus whereas the other is close to the C-terminus (*Figure 2—figure supplement 1*). The latter cysteine residue is often neighbored by glycine residues. A similar pattern is seen in SoxY proteins where the cysteine of the carboxy-terminal GGCGG sequence has a well-established sulfur substrate-binding function. Sulfur-loaded SoxY serves as a substrate for the other components of the periplasmic thiosulfate oxidizing Sox multienzyme complex (*Sauvé et al., 2007*).

Phylogenetic analysis of LbpA proteins encoded in group I gene clusters (*Figure 3*) clearly showed two distinct types of LbpA proteins, LbpA1 and LbpA2. In all organisms containing two *lbpA* genes, one gene encodes an LbpA1 and the other a member of the LbpA2 type (*Figure 3*). When the analysis was widened to all LbpAs and representative *bona fide* GcvH proteins, the LbpA1 and LbpA2 differentiation for the proteobacterial proteins and those from Aquificae remained well supported. Deep branching points, such as those for the GcvH proteins or the LbpA proteins from Actinobacteria, Firmicutes and Archaea however lacked significance (*Figure 3—figure supplement 1*).

## Production and analysis of LbpA1 and LbpA2 proteins in vivo and in vitro

Four different LbpA proteins were produced in *E. coli* (*Table 1*). Proteins TK90_0638 (Tk90LbpA1) and TK90_0640 (TkLbpA2) are representatives of the LbpA1 and LbpA2 groups, respectively and stem from the same gammaproteobacterial host, the obligately chemolithoautotrophic alkaliphilic sulfur oxidizer *Thioalkalivibrio* sp. K90mix. To allow generalization, one further protein from each group was chosen from organisms that contain only one *lbpA* in their *hdr*-like gene cluster. Protein ThisiDRAFT_1533 (TsLbpA1) from the purple sulfur bacterium *Thiorhodospira sibirica*, a member of the Gammaproteobacteria, served as another example from the LbpA1 group whereas Hden_0696 (HdLbpA2) from *H. denitrificans* is a representative of the LbpA2 proteins (*Figure 3—figure supplement 1*). TsLbpA1 and HdLbpA2 were analyzed by analytical gel permeation chromatography and eluted as monomers of the expected sizes (*Figure 4—figure supplement 1*). *Thiorhodospira* TsLbpA1 was analyzed by mass spectrometry and yielded a value 17,457 Da which matched that of the unmodified apoprotein.

Carboxy-terminally Strep-tagged LbpA2 from *H. denitrificans* was also produced in *E. coli* Δ*iscR* from plasmid pBBR1p264HdHdrTet at slow growth rates under anoxic conditions and at low temperature (16°C). Under such conditions, maturation of recombinant proteins containing complex cofactors/prosthetic groups like iron-sulfur clusters can sometimes be improved in a suitable genetic background (*Akhtar and Jones, 2008*). Indeed, pure LbpA2 protein produced under these conditions migrated significantly faster than the unmodified apoprotein obtained by isopropyl-β-D-thiogalactopyranoside (IPTG)-induced expression from a pET-based plasmid (*Figure 4*, left panel). The low copy number plasmid pBBR1p264HdHdrTet contains the *H. denitrificans hdr*-like gene cluster from *dsrE* to *lbpA* (*Figure 2*) under a strong constitutive promoter from *Gluconobacter oxydans* (*Kallnik et al., 2010*). In *H. denitrificans* cell extracts, the fast migrating band was clearly identified as being lipoylated because it was detected by an antibody directed specifically against HdLbpA2 as well as by an anti-lipoic acid antibody (*Figure 4*, right panel). A similar behavior upon posttranslational modification has been observed for acyl carrier protein (ACP) where acyl-ACP products migrate faster than unacylated ACP on SDS-PAGE. In both cases, this property can be attributed to enhanced binding of SDS to the acidic ACP/LbpA2 molecules upon fatty acylation/lipoylation (*Shen et al., 1992*).

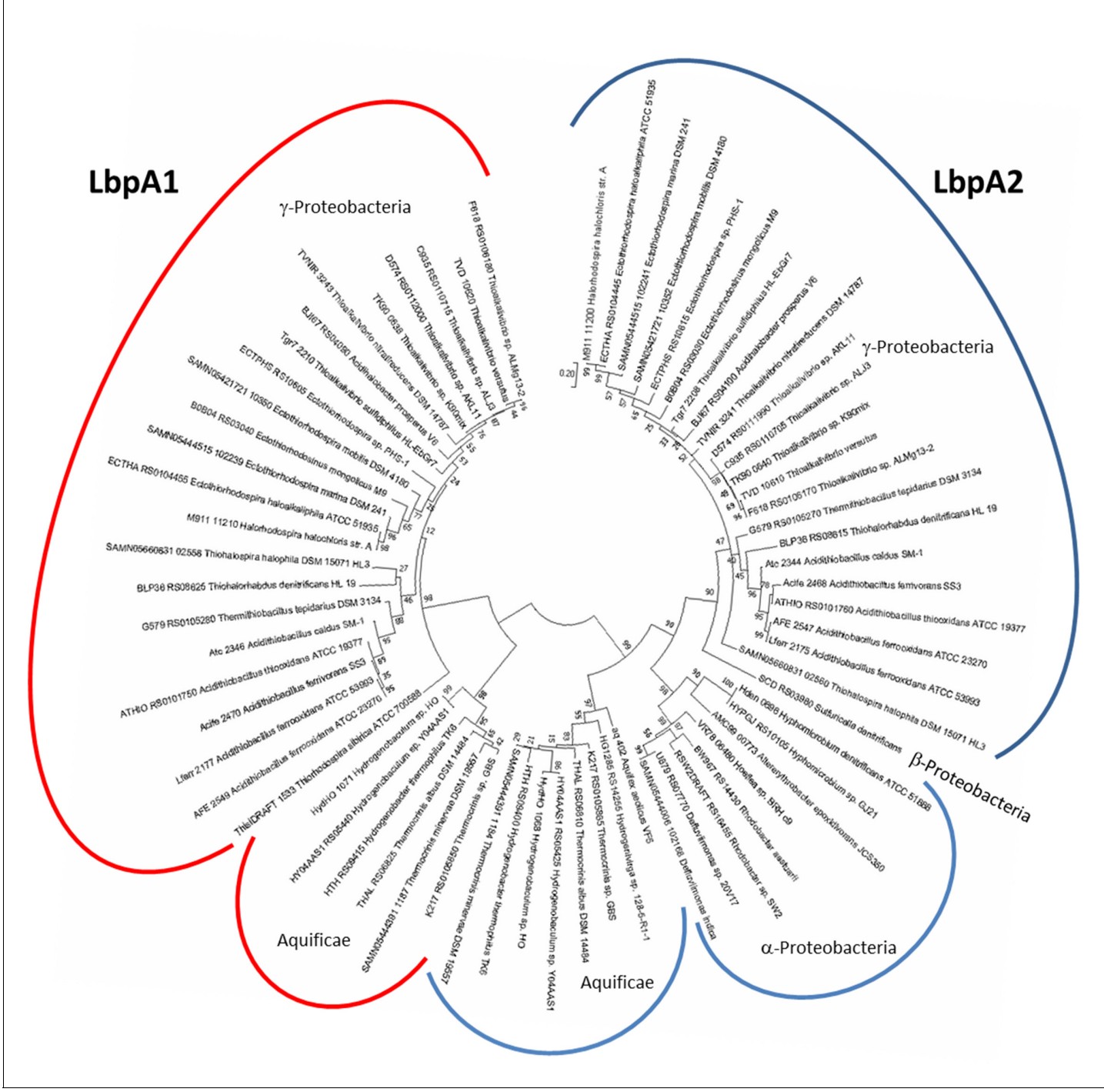

**Figure 3.** Molecular phylogenetic analysis of LbpA proteins by maximum likelihood method. First, the best amino acid substitution model was calculated in MEGA7 (**Kumar et al., 2016**). The Le_Gascuel_2008 model (**Le and Gascuel, 2008**) had the lowest BIC (Bayesian information Criterion) score and was considered to describe the substitution pattern the best. The evolutionary history was then inferred by using the Maximum Likelihood method based on this model. The tree with the highest log likelihood (−5373.74) is shown. Bootstrap values given at branching points are based on 1000 replicates. Initial tree(s) for the heuristic search were obtained automatically by applying Neighbor-Join and BioNJ algorithms to a matrix of pairwise distances estimated using a JTT model, and then selecting the topology with superior log likelihood value. A discrete Gamma distribution was used to model evolutionary rate differences among sites (5 categories (+*G*, parameter = 0.8540)). The tree is drawn to scale, with branch lengths measured in the number of substitutions per site. The analysis involved 68 amino acid sequences. All positions containing gaps and missing data were eliminated. There were a total of 123 positions in the final alingment. Evolutionary analyses were conducted in MEGA7 (**Kumar et al., 2016**).
DOI: https://doi.org/10.7554/eLife.37439.006

*Figure 3 continued on next page*

*Figure 3 continued*

The following source data and figure supplement are available for figure 3:

**Source data 1.** List of source organisms, locus tags and accessions for the LbpA and GvcH proteins considered in phylogenetic trees shown in *Figure 3* and *Figure 3—figure supplement 1*.

DOI: https://doi.org/10.7554/eLife.37439.008

**Figure supplement 1.** Molecular phylogenetic analysis of LbpA and GvcH proteins by maximum likelihood method.

DOI: https://doi.org/10.7554/eLife.37439.007

## LbpA2 is an essential component of the *Hyphomicrobium denitrificans* sulfur-oxidizing system

With very few exceptions *hdr-lbpA* containing organisms are lithotrophs that obligately depend on reduced sulfur compounds as sources of electrons for chemo- or photoautotrophic growth (Figure 1—source data 1). *H. denitrificans* $X^T$ (ATCC 51888) presents a notable exception. This ubiquitous, appendaged, budding bacterium can switch between growth on $C_1$- or $C_2$-compounds like methanol or dimethyamine and consumption of the volatile organosulfur compound DMS. The sulfur contained in the latter is fully oxidized to sulfate. Thiosulfate is an intermediate in the oxidation process and when supplied externally it can be used as an additional source of electrons during chemoorganoheterotrophic growth (*Koch and Dahl, 2018*).

Comparative proteomics indicated specific formation of proteins encoded in the *hdr-lbpA2* gene cluster during growth on DMS (*Koch and Dahl, 2018*). Western blotting using two different antibodies, one directed against *H. denitrificans* LbpA2 and the other against lipoic acid, verified this finding and furthermore proved the presence of a lipoylated protein of the expected size in cells grown on methylamine/thiosulfate but its absence in cells grown on the organic compound alone (*Figure 4*).

These findings prompted construction of a *H. denitrificans* mutant strain carrying an in-frame deletion of the *lbpA2* gene, leaving all other genes in the *hdr-lbpA2* locus intact. Phenotypic analyses unambiguously revealed that the Δ*lpbA* strain was incapable to grow on or degrade dimethylsulfide (*Figure 5A*). The phenotype was almost exactly the same as that observed for a mutant with an insertional mutation affecting *hdrA* and all genes located downstream (strain Δ*hdr* (*Koch and Dahl, 2018*), *Figure 5A*). In addition, the *lbpA2*-deficient *H. denitrificans* strain showed a clear phenotype when thiosulfate was provided as an additional electron source during chemoorganoheterotrophic growth on methanol (*Figure 5B*). While the wild type produced sulfate from thiosulfate (*Figure 5B*, middle panel), *H. denitrificans* Δ*lbpA2* formed tetrathionate, just as observed for *hdrA* insertional mutant (*Koch and Dahl, 2018*). Tetrathionate, a dead end product and nonmetabolized, is formed by a side pathway catalyzed by the thiosulfate dehydrogenase, TsdA (*Koch and Dahl, 2018*). The lower growth rate of the *H. denitrificans* wildtype compared to the Δ*hdr* and Δ*lbpA2* mutant strains on methanol and thiosulfate is most probably due to induced production of the very complex heterodisulfide reductase-like proteins in the wildtype that does not take place in the mutant strains. The growth yield increase expected for the wildtype on thiosulfate due to availability of additional electrons for respiratory energy conservation is not readily observable in *Figure 5B*. It is important to note in this respect that previous work on *Hyphomcirobium* strain EG also proved batch culture experiments unsuitable for demonstration of reproducible yield increases from thiosulfate. However, chemostat experiments unambiguously demonstrated yield increases by thiosulfate addition for *Hyphomicrobium* species (*Suylen et al., 1986*). In summary, our findings demonstrate that the GcvH-like LbpA2 protein is an indispensable component of the Hdr-like pathway of sulfur oxidation.

**Table 1.** LbpA proteins studied in this work and their identity/similarity with *E. coli* GcvH.

| Protein and locus tag | Source organism | *E. coli* GcvH (identity) | *E. coli* GcvH (similarity) |
|---|---|---|---|
| LbpA1 ThisiDRAFT_1533 | *Thiorhodospira sibirica* ATCC 700588$^T$ | 25.6% | 41.7% |
| LbpA1 TK90_0638 | *Thioalkalivibrio* sp. K90mix | 29.5% | 45.6% |
| LbpA2 TK90_0640 | *Thioalkalivibrio* sp. K90mix | 34.5% | 52.4% |
| LbpA2 Hden_0696 | *Hyphomicrobium denitrificans* ATCC 51888$^T$ | 28.8% | 45.2% |

DOI: https://doi.org/10.7554/eLife.37439.009

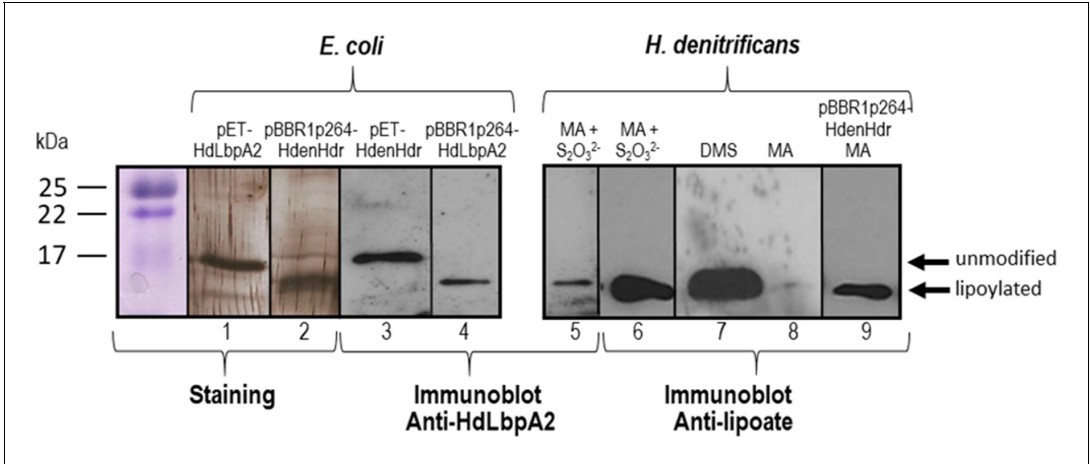

**Figure 4.** SDS-PAGE and Western blot of HdLbpA2. Lanes 1, 2, 3 and 4 were loaded with 20 ng of pure recombinant HdLbpA2 stemming from pET- or pBBR1p264-based expression as indicated. Lanes 5 to 8 were loaded with extracts (15–20 µg protein) *H. denitrificans* wildtype grown on methylamine plus thiosulfate (MA + $S_2O_3^{2-}$), dimethylsulfide (DMS) and methylamine (MA). Lane 9 was loaded with extract (17.5 µg protein) of *H. denitrificans* expressing plasmid-encoded genes *dsrEhdrC1B1AhyphdrC2B2hyplbpA* from a constitutive promoter. Lanes 1 and 2 show silver stained gels. Western blots were developed with antiserum against HdLbpA2 or with anti-lipoic acid antibody as indicated.

DOI: https://doi.org/10.7554/eLife.37439.010

The following figure supplement is available for figure 4:

**Figure supplement 1.** Purification of LbpA and LplA-like proteins and analysis of *T. sibirica* LbpA1 by gel permeation chromatography on Superdex 75.
DOI: https://doi.org/10.7554/eLife.37439.011

## LbpA1 and LbpA2 cannot replace *Bacillus subtilis* GcvH in the lipoate transfer pathway and failed to be modified by LipM

In the next step, we set out to gain insight into the specificity of the novel LbpA proteins and tested whether they are restricted to a function in oxidative sulfur metabolism or can replace related GcvH proteins in vivo. In *Bacillus subtilis,* the glycine cleavage system protein GcvH plays a central role in the modification biosynthesis of other lipoic acid requiring enzymes (*Figure 6A*). The protein:lipoate ligase of this organism is LplJ and has 33% identity with *E. coli* LplA (*Martin et al., 2011*). In addition, a single domain LplA family protein, LipM, is present that acts specifically as an ACP:GcvH octanoyl transferase (*Christensen and Cronan, 2010*; *Christensen et al., 2011*) (*Figure 6A*). LipM exclusively modifies the *B. subtilis* GcvH (*Figure 6A*) but not other lipoic acid requiring enzymes. The modified GvcH protein then acts as a donor in a reaction catalyzed by LipL, a GvcH:[lipoyl domain] amidotransferase which transfers the octanoyl group from GcvH to the lipoyl domains (LD) of the dehydrogenases (*Martin et al., 2011*). The lipoyl sulfur atoms are finally inserted by the action of LipA. In *B. subtilis* strain NM20, disruption of *gcvH* resulted in complete loss of biosynthetic lipoyl assembly pathway and the Δ*gcvH* strain became a lipoate auxotroph (*Christensen et al., 2011*). In our previous work, we showed that diverse GcvH proteins from various organisms (including human) could successfully substitute for *B. subtilis* GcvH (*Cao et al., 2018*). Here, we attempted to complement strain NM20 with ectopic copies of LbpA1 from *T. sibirica* and LbpA2 from *H. denitrificans*. The *lbpA1* and *lbpA2* genes were integrated into the chromosomal *amyE* site of *B. subtilis* Δ*gcvH* strain under the IPTG-dependent promoter, P*spac*. No discrete colony formation was observed even with induction of IPTG indicating that neither LbpA1 nor LbpA2 could restore growth of the *B. subtilis* Δ*gcvH* strain (*Figure 6B*).

The inability of both novel proteins to complement the *B. subtilis* Δ*gcvH* strain could be due to a failure to accept octanoate in the reaction catalyzed by LipM or to an inability of LipL to transfer lipoate from the lipoate-binding proteins to the lipoyl-domains of other lipoate requiring proteins by LipL. To investigate these possibilities, we assayed LipM modification in vitro using purified *T. sibirica* LbpA1 and *H. denitrificans* LpbA2 as substrates. The *B. subtilis* GcvH protein was used as positive control. Each protein was tested for its ability to accept [1-$^{14}$C]octanoate transferred by LipM. The reactions were analyzed by autoradiography on SDS-PAGE gel (*Figure 6D*). No octanoylated LbpA1

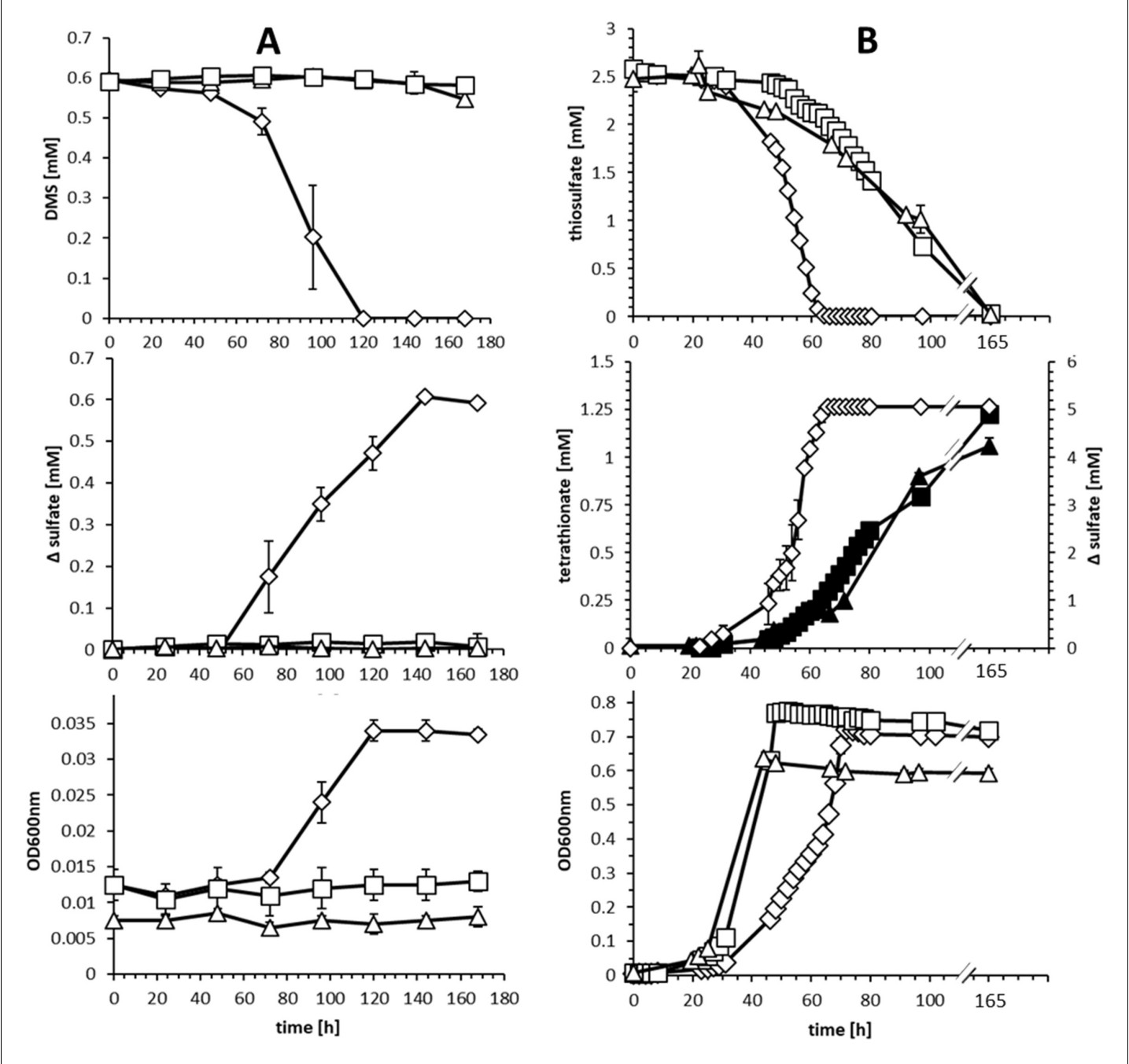

**Figure 5.** Comparison of substrate consumption, product formation and growth of wildtype and knockout strains of *H. denitrificans* during growth on different sulfur compounds. (**A**) Consumption of DMS (upper panel) and growth on DMS (lower panel) by *H.denitrificans* wt (◊) and the Δ*hdr* (□) and Δ*lbpA* (Δ) strains. DMS concentrations above 1.5 mM are toxic in batch culture and appreciable cell yields necessitate repeated feeding of cultures (***Koch and Dahl, 2018***). Stoichiometric production of sulfate from DMS for the wt is shown in the middle panel. (**B**) Consumption of thiosulfate (upper panel) on methanol-containing medium by *H. denitrificans* wt (◊), the Δ*hdr* (□) and the Δ*lbpA* (Δ) strain. Sulfate formation by the wt (open diamonds, ◊) and tetrathionate formation by the Δ*hdr* (filled boxes, ■) and Δ*lbpA* (filled triangles ▲) strains is given in the middle panel. The lower panel shows growth on methanol plus thiosulfate. Representative experiments averaged from two independent biological replicates are shown. Error bars indicate SD.

DOI: https://doi.org/10.7554/eLife.37439.012

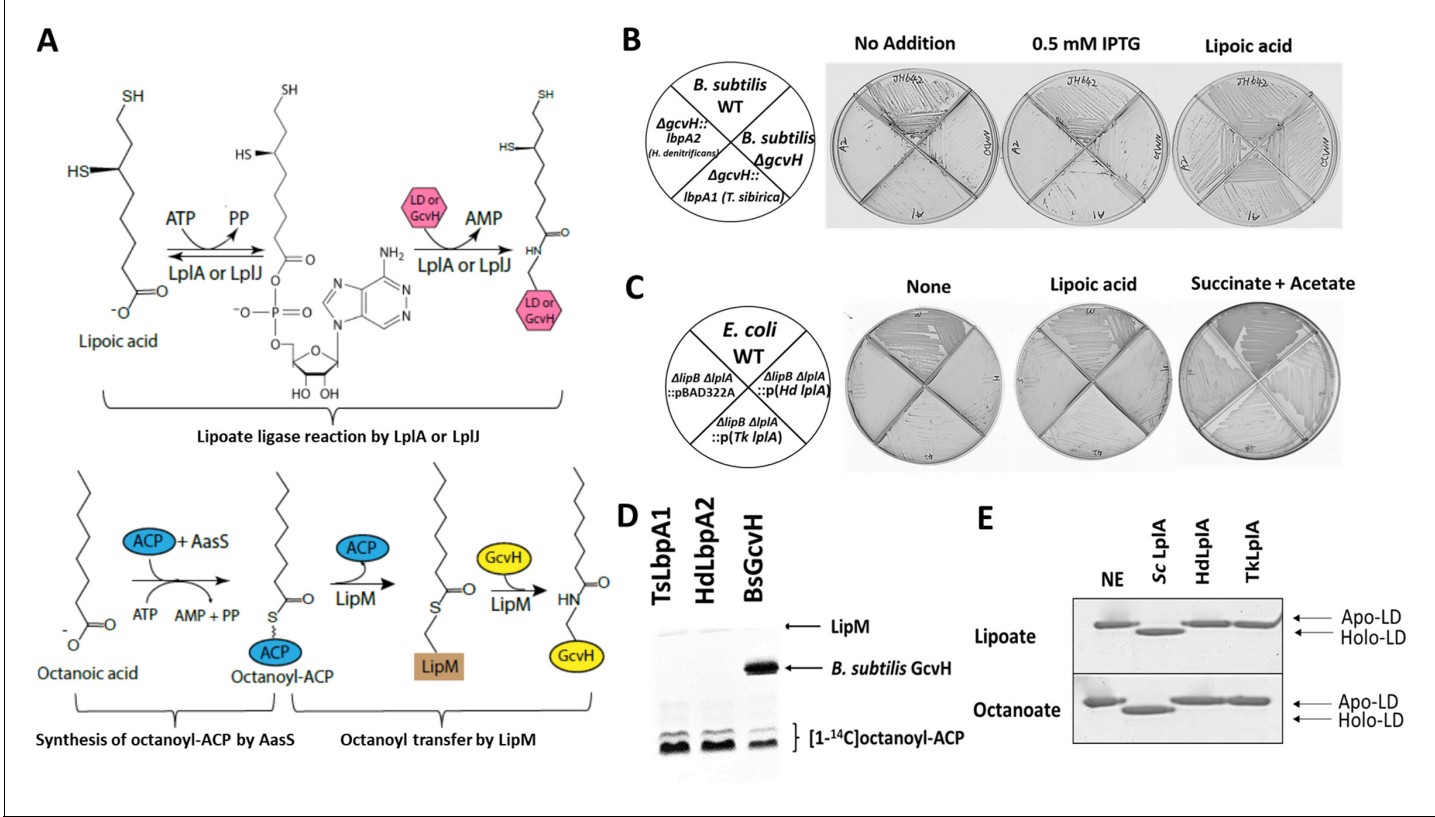

**Figure 6.** In vivo and in vitro activities of *lbpA* and LbpA. (A) Biochemical reactions for lipoate/octanoate assembly. Upper panel, lipoate ligase reaction catalyzed by LplA or LplJ. When lipoic acid is provided in the environment, LplA/LplJ catalyzes the ligase reaction in two steps. First, lipoic acid is activated to lipoyl-AMP with concomitant release of pyrophosphate. In the presence of a LD or GcvH acceptor protein, the lipoyl moiety is transferred to the LD or GcvH. LD, lipoyl domain. Lower panel, synthesis of octanoyl-ACP and amidotransfer of octanoyl moiety from ACP to GcvH. The synthesis of octanoyl-ACP requires AasS (the acyl-ACP synthetase from *Vibrio harveyi*) *holo*-ACP, ATP and free octanoic acid. The octanoyl transfer step requires LipM, which is the octanoyl transferase in *B. subtilis*. (B) Complementation of *B. subtilis* Δ*gcvH* NM20 strain with *lbpA* genes from sulfur oxidizers and LipM modification assay. The *lbpA* genes were integrated into the chromosomal *amyE* site of the *B. subtilis* Δ*gcvH* strain under control of P*spac* promoter and were inducible by IPTG. Colony formation was scored on minimal agar plates with glucose as the sole carbon source and supplements indicated after 36 hr at 37°C. The wild type (JH642) and *B. subtilis* Δ*gcvH* strain (NM20) served as positive and negative controls, respectively. No discrete colonies were formed by the strains that expressed LbpA1 and LbpA2 (the imperfections are scratches from the streaking), whereas strains all grew well with lipoic acid supplementation. (C) Complementation of lipoate auxotrophic *E. coli* Δ*lipB* Δ*lplA* strain with LplA proteins from sulfur oxidizers. The control strains were the wild-type (WT) strain and *E. coli* Δ*lipB* Δ*lplA* strain containing the empty vector (pBAD22A). Strains were grown on M9 minimum glycerol media plates with the indicated supplements at 37°C for 36 hr. Complementation proceeded both in the presence and absence of arabinose induction. Plates above contained 0.2% arabinose. (D) Assay of [1-$^{14}$C]octanoyl transfer from [1-$^{14}$C]octanoyl-ACP to lipoate-binding proteins (TsLbpA1 from *T. sibirica* and HdLbpA2 from *H. denitrificans*) and to *B. subtilis* GcvH (BsGcvH) as indicated in the figure. The [1-$^{14}$C]octanoyl-ACP was synthesized in situ by *Vibrio harveyi* AasS acyl-ACP synthetase (ATP was added for the AasS reaction). Reaction was analyzed on SDS-PAGE by autoradiography. (E) Mobility shift assay for analysis of in vitro lipoylation and octanoylation catalyzed by LplA using the E2$_{AceF}$ *E. coli* lipoyl domain (LD) as acceptor. Loss of the positive charge of the modified lysineε—amino group of the LD results in faster migration of the modified form on Urea-PAGE gels. NE, no enzyme; Sc LplA, LplA from *S. coelicolor*; HdLplA, LplA from *H. denitrificans*; TkLplA, LplA from *Thioalkalivibrio* sp. K90mix.

DOI: https://doi.org/10.7554/eLife.37439.013

or LbpA2 band was observed suggesting the failure of in vivo complementation was due to lack of modification by LipM. Hence we conclude that neither of the novel lipoate-binding proteins is able to functionally replace the *B. subtilis* GcvH protein as a hub in octanoyl group transfer.

## The LplA homologs from sulfur oxidizers cannot replace LplA or LipB in *E. coli*

To determine whether the LplA lipoate-protein ligase homologs from *H. denitrificans* (HdLplA, Hden_0686, *Table 2*) and the *Thioalkalivibrio* sp. K90mix (TkLplA, TK90_0642, *Table 2*) have LplA

**Table 2.** Protein identity/similarity between LplA-like proteins studied in this work and their identity/similarity with *E. coli* LplA, LipB and *B. subtilis* LipM respectively.

| Locus tag | Source organism | *E. coli* LplA (Identity/Similarity) | *E. coli* LipB (Identity/Similarity) | *B. subtilis* LipM (Identity/Similarity) |
|---|---|---|---|---|
| Hden_0686 | *Hyphomicrobium denitrificans* ATCC 51888[T] | 18.8%/33.1% | 10.8%/18.5% | 16.9%/31.8% |
| TK90_0642 | *Thioalkalivibrio* sp. K90mix | 23.9%/36% | 10.4%/19.2% | 18.9%/33.6% |

DOI: https://doi.org/10.7554/eLife.37439.014

activity and may be involved in maturation of the novel LbpA proteins from sulfur oxidizers, we first tested their ability to restore growth of the *E. coli* Δ*lipB* Δ*lplA* strain QC146 (*Christensen and Cronan, 2009*). The Δ*lipB* and Δ*lplA* deletions of this strain result in an inability to synthesize lipoate and to scavenge lipoic acid from the medium. Complementation was tested on M9 minimal agar plates using glycerol as the sole carbon source to avoid bypass of succinate- and acetate-dependent growth by fermentative metabolism (*Herbert and Guest, 1968*). Due to the *E. coli* QC146 Δ*lplA* mutation, the strain is unable to grow with lipoic acid supplementation. However, growth proceeds robustly when a plasmid encoding lipoate ligase activity is present (*Cao and Cronan, 2015*; *Christensen and Cronan, 2009*). Upon expression of the putative *lplA*-encoding genes from sulfur compound oxidizers, the *E. coli* strain failed to grow, either in the presence or in the absence of lipoic acid. The strains all grew well upon addition of succinate and acetate which bypass the requirement for a cellular lipoylation pathway (*Figure 6C*). This indicates that the proteins cannot serve as lipoate ligases (at least not for *E. coli* lipoate cognate proteins) in the presence of free lipoate and that they fail to act as octanoyl transferases in the *E. coli* background (data not shown).

To confirm the in vivo complementation results, in vitro lipoylation assays with *E. coli* LD (E2$_{AceF}$) as acceptor protein were performed with purified HdLplA and TkLplA (*Figure 4—figure supplement 1*). *Streptomyces coelicolor* LplA, was used as the positive control (*Cao and Cronan, 2015*). We found that, *E. coli* LD was readily lipoylated in the presence of lipoic acid and ATP by *S. coelicolor* LplA as shown by the faster migration on native gel electrophoresis due to loss of the positive lysine charge upon modification (*Hermes and Cronan, 2009*). However, LplA from *H. denitrificans* and *Thioalkalivibrio* sp. K90mix had no activity with *E. coli* LD (*Figure 6E*, top panel). The same result was observed with octanoate, another in vivo and in vitro substrate of *E. coli* LplA (*Morris et al., 1994*) (*Figure 6E*, bottom panel). These in vitro results agree with the finding that both proteins fail to complement an *E. coli* strain devoid of LplA and LipB in in vivo complementation assays.

## LplA-like protein from sulfur oxidizing bacteria is a *bona fide* lipoate-protein ligase active on its own lipoate-binding proteins

Given our findings that the novel LbpA proteins from sulfur oxidizers could not be modified by other lipoate metabolism proteins and that in turn LplA-like proteins from sulfur oxidizing bacteria fail to show lipoylation activity with the *E. coli* cognate proteins, we hypothesized that LplA-like proteins encoded in *hdr*-like gene cluster have activity only upon their natural cognate proteins, which are the LbpA proteins encoded by the closely linked genes.

To test this hypothesis, we purified LbpA1, LbpA2 and LplA-like proteins from a single organism *Thioalkalivibrio* sp. K90mix for the purpose of consistency and performed a series of in vitro analyses. First, we assayed the LplA-like protein for both the overall ligase reaction and the first partial reaction (activation of lipoic acid/octanoic acid with ATP to form lipoyl/octanoyl-AMP intermediate) of the ligase reaction. In the absence of an acceptor protein, synthesis of both octanoyl-adenylate and lipoyl-adenylate intermediates were readily demonstrated by use of ATP labeled in the α-phosphate (*Figure 7A*). Moreover, upon addition of an acceptor protein, LbpA1 or LbpA2, the adenylate intermediates were hydrolyzed to AMP. This was readily observed by the increased formation of AMP. This could be clearly observed by the increased levels of AMP. The overall lipoate ligase activity of LplA-like proteins was confirmed by mass spectrometry. Lipoylated Lbp proteins had a mass increase of about 185 Da for both Lbp proteins (184.5 Da for LbpA1 and 186.2 for LpbA2) comparing with the unmodified forms (17457.1 Da for LbpA1 and 16410.7 for LbpA2) (*Figure 7B and C*). The mass increases matched well with the expected value (188 Da) upon addition of a lipoyl group. We further determined the ligase activity of LplA-like protein by autoradiographic analysis using free [1-$^{14}$C]

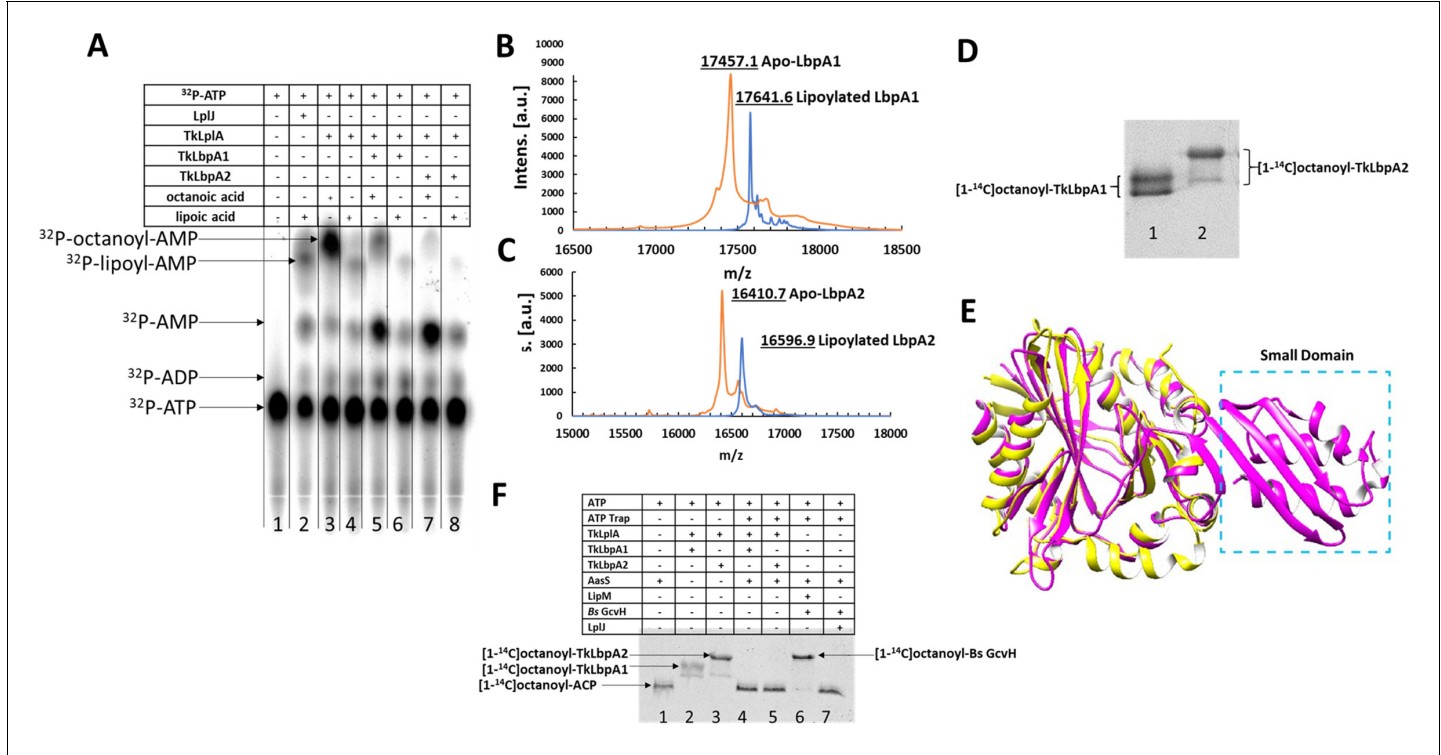

**Figure 7.** Enzymatic role of LplA from sulfur oxidizing bacteria in lipoylation/octanoylation of Lbp proteins. (A) TLC analysis of products formed from [$\alpha$-$^{32}$P]ATP. Both synthesis of lipoyl/octanoyl-5'-AMP and transfer of the lipoyl/octanoyl moiety to the acceptor proteins LbpA1 and LbpA2 are shown respectively. Addition of an acceptor protein results in consumption of the acyl adenylate intermediates and production of AMP. TkLplA, TkLbpA1 and TkLbpA2 denote LplA, LbpA1 and LbpA2 proteins from *Thioalkalivibrio* sp. K90mix, respectively. (B and C) MALDI mass spectrometric analysis of lipoylation of TkLbpA1 and TkLbpA2 respectively catalyzed by LplA from *Thioalkalivibrio* sp. K90mix. The mass of the lipoylated LbpA1 (17641.6 Da) and lipoylated LbpA2 form (16596.9 Da) agree well the calculated values. The change in mass upon modification (calculated for lipoyl modification, 188 Da; observed, 184.5 Da for LbpA1 and 186.2 Da for LbpA2 respectively) is within the accuracy of the instrument utilized. Intens., intensity; a.u., arbitrary units. (D) Activity of TkLplA (KEGG: TK90_0642) on the two lipoate-binding proteins. [1-$^{14}$C]octanoic acid was used as the substrate. Reactions were incubated at 37°C for 1 hr and loaded on a 15% SDS-PAGE gel. The [1-$^{14}$C]octanoyl-Lbp proteins were detected by autoradiography. Lane 1, LbpA1 from *Thioalkalivibrio* sp. K90mix (TK90_0638); lane 2, LbpA2 from *Thioalkalivibrio* sp. K90mix (TK90_0640). (E) Superimposition model of from *Thioalkalivibrio* sp. K90mix LplA (yellow) obtained by threading on the structure of *E. coli* LplA (magenta, PDB 3A7R). The small domain of *E. coli* LplA was denoted by blue dotted box. (F) Detection of [1-$^{14}$C]octanoyl transfer from [1-$^{14}$C]octanoyl-ACP to lipoate-binding proteins (Lbps). The [1-$^{14}$C] octanoyl-ACP was synthesized in situ from [1-$^{14}$C]octanoate, *V. harveyi* AasS acyl ACP synthetase, ATP and *E. coli* holo-ACP, whereas formation, if any, of [1-$^{14}$C]octanoyl-Lbp should require apo-Lbp, octanoyl transferase and [1-$^{14}$C]octanoyl-ACP. Note that the residual ATP remaining from the AasS reaction was depleted by use of an ATP trap (2 units of hexokinase plus 10 mM D-glucose to convert ATP to glucose-6-phosphate and ADP). Each reaction contained ATP, [1-$^{14}$C]octanoate, and *E. coli* holo-ACP. Lane 1, [1-$^{14}$C]octanoyl-ACP; Lane 2, positive control LbpA1 (TK90_0638) in the presence of ATP and LplA (TK90_0642); Lane 3, positive control LbpA2 (TK90_0640) in the presence of ATP and LplA (TK90_0642); Lane 4, LbpA1with AasS coupled reaction in the presence of LplA and ATP trap; Lane 5, LbpA2 with AasS coupled reaction in the presence of LplA and ATP trap; Lane 6, *B. subtilis* GcvH protein with AasS coupled reaction in the presence of a real octanoyltransferase LipM from *B. subtilis* and ATP trap.

DOI: https://doi.org/10.7554/eLife.37439.015

The following figure supplement is available for figure 7:

**Figure supplement 1.** Mobility shift assay for analysis of in vitro octanoylation catalyzed by HdLplA from *Hyphomicrobium denitrificans* (Hden_0686) using LbpA1 from *Thiorhodospira sibirica* (TsLbpA1, ThisiDRAFT_1533) as acceptor.

DOI: https://doi.org/10.7554/eLife.37439.016

octanoate as a substrate (*Figure 7D*). The reaction was also confirmed by MALDI-mass spectral analysis and the mass difference between *apo* and *holo*-Lpb was exactly the mass of the octanoyl addition (data not shown). Note that the purified LbpA1 and LbpA2 proteins migrated as doublets when analyzed on SDS/PAGE gels (*Figure 7D*), which is attributed to deamidation during purification and does not affect modification (*Jordan and Cronan, 1997*; *Robinson and Robinson, 2001*). Gel mobility shift assays showed that LbpA and LplA-like proteins encoded in or close to *hdr*-like gene clusters exhibit cross-species functionality (*Figure 7—figure supplement 1*).

Lipoate-protein ligases belong to the Pfam PF03099 protein ligase family. This group of proteins are constructed on the same scaffold albeit they have markedly different sequences and include both classical ligases and other enzymes catalyzing acyl transfer, such as amidotransferases and octanoyltransferases (*Cronan, 2016*). The classical lipoate ligases seem to have a consistent overall architecture: a large catalytic domain and a small auxiliary domain at either the C-terminus of the protein such as *E. coli* LplA, at the N-terminus such as *S. coelicolor* LplA (*Cao and Cronan, 2015*) or as a small accessory domain encoded by a separate gene, such as in *Thermoplasma acidophilum* LplB (*Christensen and Cronan, 2010*; *McManus et al., 2006*; *Posner et al., 2013*). Upon threading the lipoate ligase of *Thioalkalivibrio* sp. K90mix onto the known *E. coli* LplA structure (PDB: 3A7R), we found to our surprise that the LplA-like protein lacked the small domain and only had a single large domain, which aligns very well with the large catalytic domain of *E. coli* LplA (*Figure 7E*). The same situation applies to LplA-like proteins from other sulfur-oxidizing bacteria (data not shown). Hence, these enzymes provide an exception to the canonical rule that a conventional lipoate-protein ligase should have two domains in order to catalyze the two partial reactions (activation of lipoic acid to lipoyl-AMP and transferring the lipoate moiety from lipoyl-AMP to its cognate proteins).

## LplA-like protein from sulfur-oxidizing bacteria does not exhibit octanoyltransferase activity

Octanyol transferases are not able to use free octanoic or lipoic acid as substrates but catalyze the ATP-independent transfer of octanoyl groups from donor proteins (acyl carrier protein or *B. subtilis* GcvH) to acceptor proteins. In spite of the different reactions catalyzed (ligase versus transferase), the octanoyl transferases *E. coli* LipB and *B. subtilis* LipM of lipoyl assembly also belong to the Pfam PF03099 family. LipB as well as LipM lack the small domain of the ligases and have a single-domain structure, just as the LplA-like protein from sulfur oxidizers. Therefore, we tested whether *Thioalkalivibrio* LplA-like protein could also have octanoyl transferase activity. That is, the protein might function in both lipoate scavenge and de novo lipoyl assembly. To test function in lipoyl assembly, we performed octanoyl transferase reactions coupled to AasS (acyl-ACP synthetase from *V. harveyi*) (*Figure 7F*). Since any residual ATP left from the AasS reaction would have complicated analysis it was removed by adding an ATP trap (hexokinase plus D-glucose), which converted any ATP to glucose-6-phosphate plus ADP (*Olsen et al., 1989*). After autoradiographic analysis, we found that the LplA-like protein from the sulfur oxidizer lacked octanoyl transferase activity regardless of whether LbpA1 or LbpA2 were present as acceptor proteins. Therefore, the *Thioalkalivibrio* LplA-like protein has only lipoate (or octanote) scavenge activity and plays no role in de novo lipoyl assembly.

## Discussion

Although the existence of lipoic acid has been known for more than sixty years (*Cronan, 2016*), our work indicates not only an unexpected metabolic function for this cofactor in biology but also shows that the mechanisms by which it is synthesized and becomes attached to protein are even more diverse than currently known.

The universal occurrence of *lbpA* in all *hdr*-like gene clusters (*Figure 2*, *Figure 2—source data 1*) in conjunction with the genetic evidence presented, establish the lipoate-binding proteins discovered in this study as key components in the novel Hdr-like pathway of sulfur oxidation. In most prokaryotes specialized for lithotrophic growth on reduced inorganic sulfur compounds like sulfide or thiosulfate, the initial steps occur outside of the cytoplasm (*Dahl, 2015*; *Venceslau et al., 2014*). These include the release of the more oxidized sulfone sulfur atom in thiosulfate as sulfate and/or the formation of polysulfide. The same holds true for *H. denitrificans* where sulfide is released from DMS and further processing is initiated in the periplasm (*Eyice et al., 2018*; *Koch and Dahl, 2018*). Subsequent oxidative reactions occur after transfer of the sulfur into the cytoplasm where it is processed in a protein-bound persulfidic state (*Figure 8*). Rhodanese- or DsrE-like sulfurtransferases as well as the sulfur carrier protein TusA appear to be of ubiquitous relevance for delivery of sulfane sulfur into the different enzymatic machineries generating sulfite, which is then usually further oxidized to sulfate (*Dahl, 2015*). Details of the sulfur oxidation route employing Hdr-like proteins are just emerging. The similarity of its components with subunits of archaeal heterodisulfide reductase points at (protein-bound) persulfide and/or disulfides (RSS⁻, RSSR) as possible substrates/intermediates in the reaction cycle. The LpbA protein is a prime candidate as a sulfur substrate-binding entity

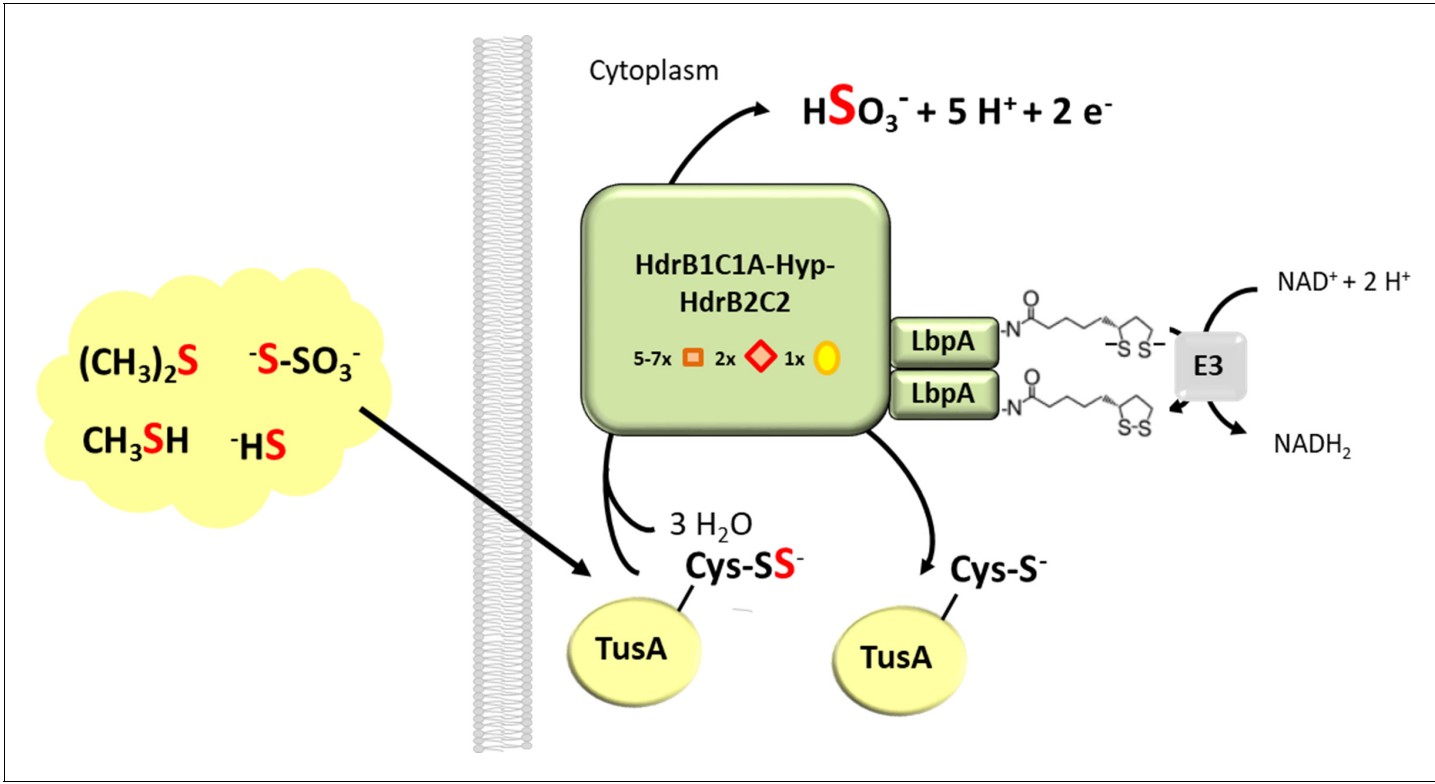

**Figure 8.** Proposed function of LbpA proteins in the Hdr-like pathway of sulfur oxidation. The TusA protein is proposed as a central sulfur carrier in the cytoplasm and collects sulfur stemming from the oxidation of organic and reduced inorganic sulfur compounds like DMS, methanethiol, sulfide or thiosulfate (*Dahl, 2015*; *Koch and Dahl, 2018*). A complex of Hdr-like proteins containing non-cubane [4Fe-4S] clusters (red diamonds) that serve as the (hetero)disulfide-reducing active site as well as regular [4Fe-4S] clusters (orange boxes) and FAD oxidizes the protein-bound sulfur to sulfite. In the course of the reaction two of the four electrons released per reaction cycle may serve as reductants for dihydrolipoyl formation on LpbA. Dihydrolipoyl dehydrogenase (indicated here as component E3 of the pyruvate dehydrogenase complex) then catalyzes reoxidation of the cofactor to lipoamide and concomitant reduction of NAD$^+$.

DOI: https://doi.org/10.7554/eLife.37439.017

that might present the sulfur substrate to different catalytic sites of the Hdr-like complex. This idea is corroborated by in vitro studies with bovine and one microbial rhodanese showing that dihydrolipoate can serve as sulfane sulfur acceptor (*Cianci et al., 2000*; *Silver and Kelly, 1976*; *Villarejo and Westley, 1963a*, *1963b*).

Currently, the reason for the presence of two different LbpA proteins in some sulfur oxidizers is unexplained. Sequence comparison does not indicate conserved differences between obviously functionally important amino acids like cysteine between the LbpA1 and LbpA2 types. It should be noted that the E2 subunits of pyruvate dehydrogenases contain up to three related but non-identical lipoyl domains. Deletion of one or even two of these domains as well as permutation has no adverse effect on assembly and catalytic activity and it appears that the reach of the outermost domain is extended by the other two domains residing more inwards in the enormous assembled pyruvate dehydrogenase complex. Group transfer and redox reactions are possible between lipoyl groups of E2 proteins (*Perham, 2000*). It is likely that LbpAs from sulfur oxidizers serve a similar substrate-channeling function and that more than a single LbpA monomer is present and functional in the mature Hdr-like multienzyme system. Gene duplication may have occurred allowing more efficient protein production followed by independent evolution of both genes. The reaction center subunits of phototrophic bacteria are probably the most prominent example for such an evolutionary history. In that case, an original protein homodimer encoded by one gene evolved into a heterodimer of two related yet distinct proteins after gene duplication (*Blankenship, 2010*).

With the exception of the sulfite/sulfate couple the midpoint reduction potential of reduced sulfur compounds and their oxidation products is more positive than that of the NAD$^+$/NADH couple

(*Thauer et al., 1977*). It is therefore generally put forward that the production of NADH and NADPH ultimately needed for carbon dioxide fixation in sulfur-oxidizing lithoautotrophs requires reverse electron flow from quinol onto $NAD^+$ at the expense of proton motive force. As depicted in *Figure 8*, the presence of lipoate as a key cofactor in the Hdr-like sulfur-oxidizing system bears the possibility that two of the four electrons released in the catalyzed reaction (R-S-S$^-$ + 3 $H_2O$ → R-S$^-$ + $HSO_3^-$ + 5 $H^+$) are used to generate dihydrolipoamide during the reaction cycle. Since the lipoamide/dihydrolipoamide system possesses one of the lowest standard biological redox potentials ($E^{0'}$=−0.29 V), the electrons released upon reoxidation of dihydrolipoamide can be directly transferred onto $NAD^+$. This reaction could considerably reduce the need for energy-demanding reverse electron flow in lithoautotrophic sulfur oxidizers. In fact, a gene for a *bona fide* dihydrolipoamide dehydrogenase is tightly linked with the *hdr*-gene cluster in sulfur oxidizing archaea (*Liu et al., 2014*). Such a gene is not apparent in bacterial *hdr*-like gene clusters but the function may be provided by an E3 subunit that is shared with the lipoate-dependent 2-oxoacid dehydrogenases just as the same E3 is used for pyruvate and α-ketoglutarate dehydrogenases of *E. coli* (*Steiert et al., 1990*).

The lipoate-binding proteins studied in this work are specifically designed for sulfur oxidation. Despite considerable sequence similarity, they cannot functionally replace the GcvH protein in *B. subtilis.* Our findings are supported by results of an earlier study, in which the LbpA protein from *Aquifex aeolicus* (aq_402, by then named GvcH5) also failed to complement *B. subtilis* Δ*gcvH* and could not be modified by *B. subtilis* LipM (*Cao et al., 2018*). Furthermore, this protein had no or only extremely weak glycine cleavage acticity and lacked lipoyl-relay activity in the the *Bacillus* background, that is it neither served as functional part of the glycine cleavage system nor as a substrate for lipoate transfer to other lipoic acid requiring proteins.

The LbpA proteins from sulfur oxidizers do not serve as substrates for the biosynthetic machineries involved in lipoate-binding protein maturation in *B. subtilis* and *E. coli.* On the other hand, the first biosynthetic protein characterized in vitro from the sulfur oxidizers, an LplA-like lipoate-protein ligase does not recognize lipoyl domains/proteins from organisms lacking components of a Hdr-like sulfur-oxidizing system. Such specificity is unprecedented, usually protein substrates are interchangeable in vitro (*Cao and Cronan, 2015*; *Cao et al., 2018*; *Christensen and Cronan, 2010*; *Posner et al., 2013*).

All sulfur oxidizers analyzed in this study contain the genetic equipment for lipoate-protein biosynthesis via previously established pathways (*Figure 2—source data 1*). These biosynthetic routes appear to be employed exclusively for the known lipoate-binding components of 2-oxoacid dehydrogenases and glycine cleavage systems. In the organisms studied in detail in this work, *bona fide* GcvH proteins are encoded in the vicinity of the other components of their glycine cleavage systems (Hden_2793 is *H. denitrificans* GcvH and TK90_1720 is GcvH from *Thioalkalivibrio* sp. K90mix). The LpbA proteins involved in sulfur oxidation obviously differ so much from GcvH proteins that they are not recognized by the GvcH maturation enzymes but are specifically modified by lipoate-protein ligases encoded in the same gene cluster.

Our finding that TkLplA is an active lipoyl/octanoyl ligase despite lacking the small C-terminal domain seen in all other lipoate ligases studied to date was unexpected. As mentioned above the small domain can be covalently attached to either terminus of the large domain or be a separate subunit (*Cronan, 2016*). However, the role of this domain/subunit in the ligase mechanism is unclear. Prior work showed that both domains are required for the overall reaction and for lipoyl-adenylate formation whereas only the large domain was required for transfer of lipoate from the adenylate to the acceptor protein (*Cao and Cronan, 2015*; *Christensen and Cronan, 2010*; *McManus et al., 2006*; *Posner et al., 2013*). The exact role of the small auxiliary domain remains obscure despite crystal structures of the isolated large domain, the isolated small domain and two-domain complex from the bipartite *Thermoplasma acidophilum* ligase. Apparently, all of the catalytic machinery lies in the large domain. This is where ATP binds and forms the adenylate (*Kim et al., 2005*; *Posner et al., 2013*). However, the large domain cannot synthesize lipoyl-adenylate without interaction with the small domain. This interaction requires substrate and upon association both domains undergo marked conformational changes (*Posner et al., 2013*). It seems clear that conformational changes in the large domain that occur upon binding substrate and interaction with the small domain somehow trigger the overall ligase reaction. If this is the case, how does HdLplA and TkLplA function as a ligase without a small domain? The most straightforward hypothesis is that the lipoate-

binding proteins LbpA1 and LbpA2 not only act as lipoate acceptors but also have a motif that triggers the conformational change usually engendered by the small domain. If so, this could explain the inability of either the HdLplA/TkLplA ligase or LbpA1 and LbpA2 binding proteins to interact with the ligases and acceptor domains of central metabolism proteins of other bacteria.

## Materials and methods

### Media and chemicals

*Escherichia coli* strains were grown on LB rich or M9 minimal media under aerobic conditions (*Sambrook et al., 1989*) at 37°C unless otherwise indicated. *Bacillus subtilis* strains were grown on LB medium at 30°C. *H. denitrificans* strains were cultivated in minimal media kept at pH 7.2 with 100 mM 3-(N-Morpholino)propanesulfonic acid (MOPS) buffer and containing per liter: 1 g $NH_4Cl$, 0.2 g $MgSO_4 \times 7 H_2O$, 0.5 g $NaH_2PO_4 \times 2 H_2O$, 1.55 g $K_2HPO_4$ and 0.2 ml $l^{-1}$ trace element solution (*Vishniac and Santer, 1957*). Methanol, dimethylamine (DMA), methylamine (MA) or dimethyl sulfide (DMS) were added as carbon and electron source. Unless otherwise indicated, 200 ml cultures containing 24.4 mM methanol or 50 mM methylamine were shaken in 500 ml Erlenmeyer flasks at 200 rpm and incubated at 30°C. If desired 2.5 mM thiosulfate were added. For growth on DMS, 500 ml serum vials sealed with butyl rubber stoppers containing 100 ml medium were used. *Thiorhodospira sibirica* was grown as previously described (*Bryantseva et al., 1999*). *Thioalkalivibrio* sp. K90mix cell material was kindly provided by Dimitri Sorokin. Antibiotics (Sigma Chemical) were used at the following concentrations (in µg ml$^{-1}$): for *E. coli*, sodium ampicillin, 100; for *B subtilis* spectinomycin sulfate, 100; sodium ampicillin, 100; kanamycin sulfate, 50; for *H denitrificans* kanamycin sulfate, 50; tetracycline HCl, 15, sodium ampicillin, 100, rifampin, 50; streptomycin sulfate, 200. [1-$^{14}$C]Octanoic acid was purchased from Moravek. American Radiolabeled Chemicals provided [α-$^{32}$P]ATP. The pH of buffers and solutions is reported at room temperature.

### Bacterial strains plasmids and molecular biological techniques

All strains used are listed in *Supplementary file 1*. All primers and plasmids used are listed in *Supplementary file 2*. Standard techniques for DNA manipulation and cloning were used unless otherwise indicated (*Ausubel et al., 1997*). PCR amplification was performed using Taq, Phusion or Q5 polymerase (New England Biolabs) according to the manufacturer's recommendation. Genomic DNA from *T. sibirica*, *H. denitrificans* and *Thioalkalivibrio* sp. K90mix was prepared using the First-DNA all-tissue Kit (GEN-IAL GmbH, Troisdorf, Germany) and served as the templates in PCR reactions. Genes encoding lipoate-binding proteins and potential lipoate-protein ligases were placed under the control of a phage T7 promoter. The oligonucleotides used for PCR amplification and introduction of restriction sites and Strep-tag encoding sequences if necessary are listed in *Supplementary file 2*. After digestion with the respective restriction enzymes, the PCR products were ligated into the corresponding sites of pET22b resulting in amino- or carboxy-terminally Strep-tagged or His-tagged proteins.

### Complementation of *E. coli* and *B. subtilis* lipoate auxotrophs

To test the protein ligase activity, *lplA* genes from *H. denitrificans* (Kyoto Encyclopedia of Genes and Genomes (KEGG) entry: Hden_0686) and *Thioalkalivibrio* sp. K90mix (KEGG entry TK90_0642) were cloned and expressed in pBAD plasmids under control of an arabinose-inducible promoter. The *E. coli* Δ*lplA* Δ*lipB* strain QC146 was used for complementation as previously described (*Cao and Cronan, 2015*). To prevent carryover of lipoic acid, all plasmid-carrying strains were grown for 1 day on the same medium containing 0.4% glycerol, appropriate antibiotics, 5 mM acetate, and 5 mM succinate to bypass the lipoic acid-requiring aerobic pathways. Strains were then grown overnight at 37°C on M9 minimal plates with and without supplementation with lipoic acid (1 mM). Glycerol was used as the carbon source in the presence of arabinose.

To test the function of Lbp proteins, *lbpA1* from *T. sibirica* (ThisiDRAFT_1533) and *lbpA2* from *H. denitrificans* (Hden_0696) were amplified, adding ribosome binding sites plus SphI and HindIII restriction sites and cloned into plasmid pDR111. The resulting pDR111 derivatives were linearized and each transformed into the *B. subtilis* Δ*gcvH* NM20 strain, finally leading to integration of *lbpA1* or *lbpA2* into the chromosomal *amyE* site. Transformants were selected on plates containing

spectinomycin and screened for the amyE phenotype (*Cao et al., 2018*). The strains were grown on Spizizen salts minimum medium (*Spizizen, 1958*) with glucose as the sole carbon source in the presence or absence of lipoic acid at 37°C for 36 hr. All complementation assays were repeated at least three times and yielded consistent results.

## Transformation of *H. denitrificans* by electroporation

A *H. denitrificans* culture (400 ml) grown in minimal medium containing 24.4 mM methanol was harvested during early exponential phase at an optical density at 600 nm ($OD_{600}$) of 0.3 (4000 × g, 10 min, 4°C). Cells were washed twice with ice-cold water (4000 × g, 10 min, 4°C), once with ice-cold 10% (v/v) glycerol and finally resuspended in 800 µl of 10% glycerol. 50 µl aliquots of cells were mixed with 500 ng of plasmid DNA and incubated on ice for 10 min. Electroporation was carried out in 0.1 cm gap cuvettes (Bio-Budget Technologies GmbH, Krefeld, Germany) with a Bio-Rad gene pulser II (Bio-Rad Laboratories) with the following electrical settings: 2.4 kV and 200 Ω at a capacitance of 25 µF. After electroporation, 1 ml of minimal medium containing 24.4 mM methanol was added to the cuvette. Cells were transferred to an Eppendorf tube and incubated at 30°C for 6 hr. Transformants were selected by plating suitable dilutions of electroporated cells onto minimal medium agar containing 24.4 mM MeOH and the appropriate antibiotics. Plates were incubated at 30°C for up to 14 days. The resulting antibiotic resistant colonies were screened via PCR.

## Construction of *H. denitrificans* Δ*lbpA*

For markerless *in frame* deletion of the *H. denitrificans lbpA* gene (Hden_0696) by splicing by overlap extension (SOE) (*Horton, 1995*), PCR fragments were constructed using primers Fwd5'_ΔlbpA, Rev5'_ΔlbpA, Fwd3'_ΔlbpA, and Rev3'_ΔlbpA (see *Supplementary file 2*). The Δ*lbpA* fragment was inserted into plasmid pk18*mobsacB* (*Schäfer et al., 1994*) using SphI and XbaI restriction sites. The SmaI-excised tetracycline cassette from pHP45Ω-Tc (*Fellay et al., 1987*) was inserted into the Klenow filled-in BglII site of the resulting plasmid pk18*mobsacB*ΔlbpA2. The final construct pk18*mobsacB*ΔlbpA2Tc was electroporated into *H. denitrificans* Sm200. Transformants were selected on minimal medium plates containing 24.4 mM methanol and the appropriate antibiotics. Single crossover recombinants were Sm$^r$ and Tc$^r$, verified by PCR screening and plated on minimal medium containing methanol and 10% sucrose. Double crossover recombinants survived in the presence of sucrose due to loss of the vector-encoded levansucrase (SacB). The genotype of double crossover recombinants was verified by PCR and Southern hybridization experiments.

## Characterization of phenotypes and quantification of sulfur species

For growth experiments on 0.6 mM DMS, cultures were inoculated to a start $OD_{600}$ of 0.01 with pre-cultures in late-exponential growth phase cultured on 24.4 mM methanol. DMS was quantified using gas chromatography (GC). 50 µl samples were taken from the headspace and injected into a GC (PerkinElmer Clarus 480, Rascon FFAP column 25m × 0.25 micron) equipped with a flame ionization detector. Measurements were conducted at a column temperature of 200°C, an injector temperature of 150°C, and a detector temperature of 250°C. $N_2$ was used as carrier gas. DMS concentrations were calculated by regression analysis based on a seven-point calibration with standard DMS solutions in minimal medium. For growth experiments on thiosulfate, media with 24.4 mM methanol and 2.5 mM thiosulfate were inoculated to a start $OD_{600}$ of 0.005 with pre-cultures in late-exponential growth phase cultured on the same medium. Thiosulfate, tetrathionate, sulfite and sulfate were determined by colorimetric, turbidometric and HPLC methods as described previously (*Dahl, 1996*; *Franz et al., 2009*). All growth experiments were repeated three to five times. Representative experiments with two biological replicates for each strain are shown.

## Overproduction, purification and preparation of recombinant proteins

LB medium (500 ml) was inoculated with 5% (v/v) of *E. coli* precultures carrying the respective pET-based plasmid and cultivated at 37°C and 180 rpm until an $OD_{600}$ of 0.6–0.8 was reached. After induction with 0.1 mM isopropyl 1-thio-*β*-D-galactopyranoside, the cells were cultivated for four more hours under the same conditions and harvested (14,000 × g, 20 min, 4°C). In the case of His-tagged proteins, pellets were resuspended in a lysis buffer containing 20 mM sodium phosphate (pH 8.0), 500 mM NaCl, and 10% glycerol and lysed by multiple passages through a French Press.

The soluble cell extract was collected and purified by nickel affinity chromatography followed by ion exchange chromatography. Protein concentrations were determined by Bradford assay. Protein purity was monitored by SDS-PAGE. The concentrated protein solutions were dialyzed overnight in dialysis buffer containing 25 mM sodium phosphate, 10% glycerol, 1 mM Tris(2-carboxyethyl)phosphine hydrochloride and 0.2 M NaCl (pH 7.5) followed by flash freezing and storage at −80°C. Strep-tagged proteins were purified according to the manufacturer's instructions.

C-terminally Strep-tagged *H. denitrificans* LbpA2 was also produced in *E. coli* BL21 (DE3) Δ*iscR*. One liter batches of lysogeny broth (LB) medium containing 100 mM MOPS buffer pH 7.4, 25 mM glucose and 2 mM iron ammonium citrate as well as ampicillin and kanamycin were inoculated with 5% (v/v) *E. coli* precultures hosting plasmid pBBR1p264HdenHdrTet (*Koch and Dahl, 2018*) and cultivated in 2 l flasks at 37°C and 180 rpm until an $OD_{600}$ of 0.4–0.6 was reached. Cultures were then moved into an anaerobic chamber (Coy Laboratory Products, Grass Lake, USA) containing 98% $N_2$ and 2% $H_2$. Cysteine (0.5 mM), and sodium fumarate (25 mM) were added. Cultures were then transferred into completely filled and tightly closed 500 ml bottles, incubated for 48–72 hr at 16°C and harvested by centrifugation (11,000 × g, 12 min). Cells were lysed by sonication in the anaerobic chamber. After removal of insoluble cell material and membranes by centrifugation (16,100 × g for 30 min at 4°C) and ultracentrifugation (145,000 × g, 3 hr, 4°C), protein was purified by Strep-Tactin affinity chromatography according to the manufacturer's instructions (IBA Lifesciences, Göttingen, Germany) followed by concentration to a final volume of less than 0.25 ml via Amicon Ultracel-3K filters (Merck Millipore, Tullagreen, Ireland). The protein was stored under anaerobic conditions at −20°C. Purity was assessed by sodium dodecyl sulfate-polyacrylamide gel electrophoresis (SDS-PAGE).

Purification of *Bacillius subtilis* octanoyltransferase LipM, *B. subtilis* 4'-phosphopantheinyl transferase Sfp, *Streptomyces coelicolor* lipoate-protein ligase LplA, *Vibrio harvey* acyl-ACP synthetase AasS and *E. coli apo*-lipoyldomain LD (E2$_{AceF}$) was performed based on previously described protocols (*Cao and Cronan, 2015*; *Christensen and Cronan, 2010*; *De Lay and Cronan, 2007*; *Jiang et al., 2006*; *Zhao et al., 2003*).

## Immunoblot analyses

Soluble fractions of whole cell extracts and pure preparations of recombinant *H. denitrificans* LbpA2 were analyzed by SDS-PAGE. Proteins were separated on 15% SDS-polyacrylamide gels and transferred to nitrocellulose blotting membranes (Amersham Protran, 0.2 µm) at 15 V for 15 min using a Transblot semi-dry transfer apparatus (BioRad). The membranes were preblocked with TBS-T buffer (100 mM Tris base, 0.9% NaCl, 0.1% Tween 20, pH 7.6) containing 5% nonfat milk powder. LpbA2 antigens were detected with an antiserum raised against oligopeptides H$_2$N-WSSVKPTLTPGAEVA-CONH$_2$, and H$_2$N-INDSLVSNPQIANQD-CONH$_2$ comprising highly immunogenic epitopes deduced from the *H. denitrificans* nucleotide sequence (Eurogentec, Liege, Belgium). The polyclonal antibodies were commercially purified by affinity chromatography on a matrix (AF-Amino TOYO, Sigma) to which the immunogenic peptides were coupled (Eurogentec) and stored at a concentration of 1.6 mg ml$^{-1}$ in PBS containing 0.01% thimerosal and 0.1% BSA. The purified antibodies were used at a 1:500 dilution. Lipoic acid was detected with a mouse monoclonal antibody raised against lipoic acid (Santa Cruz Biotechnology, Dallas, USA) in a 1:4000 dilution. In both cases, membranes were probed for 3 hr in TBS-T containing 0.5% BSA. Following incubation with a goat anti-mouse IgG antibody horseradish peroxidase (HRP) conjugate (Merck, diluted 1:5000) or a goat anti-rabbit IgG antibody HRP conjugate (Sigma, diluted 1:4000) in TBS-T with 0.5% BSA, labelled proteins were detected with the SignalFire ECL reagent system (Cell Signalling Technology, Danvers, USA).

## Preparation of *holo*-ACP

*holo*-ACP was obtained by minor modificationsa of a published procedure (*Cronan and Thomas, 2009*).

To convert any *apo*-ACP to the holo form the reaction mixture (100 µl) contained 100 mM MES (pH 6.0), 10 mM MgCl$_2$, 5 mM DTT, 500 µM lithium CoA, 200 µM *apo*-ACP and 10 µM *B. subtilis* 4'-phosphopantheinyl transferase, Sfp (*De Lay and Cronan, 2007*). The reaction was performed at 37°C for 3 hr. The mix was then incubated at 65°C for 2 hr, and the precipitates were removed by centrifugation. The remaining fractions containing *holo*-ACP were validated by conformational-

sensitive 2 M Urea-PAGE (15% acrylamide), and then combined and dialyzed against 25 mM sodium phosphate buffer (pH 7.0), 300 mM NaCl, 1 mM (tris(2-carboxyethyl)phosphine (TCEP), 10% glycerol.

## Purification of LDs

The *E. coli* E2p(1,3) hybrid LD (E2$_{AceF}$) was purified by acid treatment followed by ion exchange chromatography as described previously (*Zhao et al., 2003*). The mass of purified lipoyl domains was verified by electrospray mass spectrometry as described for AcpP (*Christensen and Cronan, 2010*). The E2$_{AceF}$ was in the unmodified form and lacked the N-terminal methionine residue. The protein was quantified as described earlier (*Christensen et al., 2011*).

## Gel shift assay for LD modification analysis

Lipoate ligase activity was assayed by observing a mobility shift of lipoate-binding proteins upon modification by native gel electrophoresis as originally described by Miles and Guest (*Miles and Guest, 1987*). Assays contained 100 mM sodium phosphate (pH 7.0), 5 mM DTT, 1 mM sodium lipoate, 1 mM MgCl$_2$, 1 mM ATP, 20 µM lipoyl domain (LD) or lipoate-binding protein and 2 µM corresponding of the respective ligase. In octanoylation assays, sodium octanoate replaced sodium lipoate. The reaction mixtures (20 µl) were incubated at 37°C for 1 hr and all the reaction products were electrophoresed on 2 M Urea-PAGE (20% acrylamide) at 120 V for 90 min. The proteins were visualized by staining with Coomassie Blue R-250.

## Assay of enzymatic [α−$^{32}$P]-lipoyl/octanoyl-AMP intermediate formation

Lipoyl/octanoyl-AMP synthesis was done using the previously described protocol with slight modification (*Cao and Cronan, 2015*). The reactions contained 50 mM sodium phosphate buffer, pH 7.8, 10 nM [α-$^{32}$P]ATP, 10 µM MgCl$_2$, 0.1 mM sodium lipoate or octanoate, 20 µM *apo*-LbpA1 (TK90_0638) or *apo*-LbpA2 (KEGG entry TK90_0640), and 2 µM LplA (TK90_0642). All the reactions were incubated for 1 hr at 37°C. A 1 µl sample of each reaction was spotted on cellulose TLC plates and developed in isobutyric acid/NH$_4$OH/water (66:1:33). The TLC plates were dried overnight, exposed to a phosphorimaging plate and visualized using a Fujifilm FLA-3000 system.

## Assay of octanoyl-transfer

The AasS coupled reaction was performed as previously described with slight modification (*Cao et al., 2018*; *Shi et al., 2016*). The mixture (25 µl) contained 100 mM sodium phosphate buffer (pH 7.2), 50 mM NaCl, 1 mM ATP, 5 mM TCEP, 2 mM MgCl$_2$, 0.25 mM [1-$^{14}$C] sodium octanoate, 50 µM *E. coli* holo-ACP, 2.5 µM of *V. harveyi* AasS acyl-ACP synthetase, 10 µM *B. subtilis* LipM or TK90LplA (TK90_0642), and 20 µM of the tested LbpA protein. The reaction was performed at 37°C for 1 hr. Note that an ATP trap (2 units hexokinase plus 10 mM D-glucose) was added to each reaction 15 min before the addition of *B. subtilis* LipM or TK90LplA to remove any remaining ATP by conversion to glucose-6-phosphate plus ADP. The products were separated using 2 M urea PAGE (20% acrylamide), then dried under vacuum at 65°C for 2 hr and exposed to preflashed Biomax XAR film (Kodak) at −80°C for 24 hr.

## Bioinformatics

BLASTP (NCBI website) was used to find homologues of Hdr-like and LbpA proteins. A phylogenetic tree was constructed from the alignment of multiple proteins followed by elucidation of the best amino acid substitution model and Maximum Likelihood analysis based on this model. Bootstrap analysis was set to 1000 repeats. All these analyses were conducted in MEGA7 (*Kumar et al., 2016*).

## Structural modeling

A model of the *Thioalkalivibrio* sp. K90mix LplA was constructed using the automated mode of SWISS_Model as previously described (*Cao et al., 2017*). The superimposition model was created by threading *Thioalkalivibrio* sp. K90mix LplA with the *E. coli* LplA crystal structure (Protein Data Bank ID 3A7R). The final image was generated by using the UCSF Chimera package (*Pettersen et al., 2004*).

## Additional information

### Funding

| Funder | Grant reference number | Author |
| --- | --- | --- |
| Deutsche Forschungsgemeinschaft | Da 351/8-1 | Christiane Dahl |
| National Institutes of Health | AI15650 | John E Cronan |

The funders had no role in study design, data collection and interpretation, or the decision to submit the work for publication.

### Author contributions

Xinyun Cao, Data curation, Formal analysis, Validation, Investigation, Visualization, Writing—original draft, Writing—review and editing; Tobias Koch, Data curation, Formal analysis, Validation, Investigation, Visualization, Methodology, Writing—review and editing; Lydia Steffens, Renate Zigann, Data curation, Formal analysis, Validation, Investigation, Visualization; Julia Finkensieper, Formal analysis, Validation, Investigation, Visualization; John E Cronan, Conceptualization, Resources, Supervision, Funding acquisition, Validation, Investigation, Project administration, Writing—review and editing; Christiane Dahl, Conceptualization, Resources, Data curation, Formal analysis, Supervision, Funding acquisition, Validation, Investigation, Visualization, Methodology, Writing—original draft, Project administration, Writing—review and editing

### Author ORCIDs

Xinyun Cao http://orcid.org/0000-0002-7346-3909
Tobias Koch http://orcid.org/0000-0003-3390-3428
John E Cronan http://orcid.org/0000-0002-7064-312X
Christiane Dahl http://orcid.org/0000-0001-8288-7546

### Decision letter and Author response

Decision letter https://doi.org/10.7554/eLife.37439.022
Author response https://doi.org/10.7554/eLife.37439.023

## Additional files

### Supplementary files

• Supplementary file 1. Table S1 – strain list. Table listing the strains and plasmids used in this study.
DOI: https://doi.org/10.7554/eLife.37439.018

• Supplementary file 2. Table S2 primer and plasmid list. Table listing the primer and plasmid names and properties used in this study.
DOI: https://doi.org/10.7554/eLife.37439.019

• Transparent reporting form
DOI: https://doi.org/10.7554/eLife.37439.020

### Data availability

All data generated or analysed during this study are included in the manuscript and supporting files. Source data files have been provided for Figures 2 and 3.

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
