## [Decision Letter]

Thank you for submitting your article "Lipoate-binding proteins and specific lipoate:protein ligases in microbial sulfur oxidation reveal a new role for an old cofactor" for consideration by *eLife*. Your article has been reviewed by two peer reviewers, and the evaluation has been overseen by a Reviewing Editor and Gisela Storz as the Senior Editor. The reviewers have opted to remain anonymous.

The reviewers have discussed the reviews with one another and the Reviewing Editor has drafted this decision to help you prepare a revised submission.

Congratulations on your high quality manuscript, which the Reviewing Editor and two expert reviewers agree should be published in *eLife* pending some straightforward revision/experimental clarification. The details of the two reviews are appended below. The primary issues we would like you to address are the following:

1) Revise the Abstract, Introduction and Discussion so that the main message is clear to the broad *eLife* audience (recall that many readers may know very little either about lipoate biochemistry or about microbial sulfur metabolic pathways): state specifically what the new function for lipoic acid is that you have discovered. Explain to naive readers why they should care about this. Rather than claiming discovery of a "novel" function, be precise about what the function is. Claims of novelty are overdone and obscure what the exciting finding is.

2) Clarify the ambiguity pointed out by reviewer 1 regarding octanoyl-transferase activity in two different parts of the text.

3) Address the specificity of the LplA protein and the role of the A1 and A2 proteins and their redundancy. This could require some additional data.

We look forward to receiving your revised submission and thank you for submitting your work to *eLife*.

*Reviewer #1:*

The manuscript focuses on an microbiologically important and biochemically interesting process: the oxidation of sulfurous compounds in bacteria that proceeds via a protein complex resembling heterodisulfide reductase. This manuscript shows that in *Hyphomicrobium denitrificans* a lipoylated-protein (LpbA2) is involved. The manuscript also clarifies the question how the lipoate cofactor is installed in LbpA1/A2-proteins (from *Thialkalivibrio*) through a specific enzyme (LplA-like protein). In general the manuscript contains a lot of high-quality data that supports the chain of evidence. Yet, there are some points that need to be addressed in my opinion.

Text/manuscript structure: The authors characterize and investigate proteins from different organisms to draw their conclusions. I wonder whether a slight restructuring of the manuscript to avoid too many "logical jumps" between the systems would be beneficial!? Definitely the authors should add a dedicated table that lists all proteins investigated in this work, as well as their percentage sequence identities to GcvH (for LbpA1/A2) or LplA and/or LipB (for LplA-like protein).

Data/interpretation: One crucial point is the functional assignment of the LplA-like protein from *Thialkalivibrio*. If I understand the authors correctly, they claim (subsection “LplA-like protein from sulfur oxidizer bacteria is a bona fide lipoate:protein ligase active on its own lipoate-binding proteins”) that the LplA-like protein is able to perform a lipoylation, as well as octanoylation. A bit later, however, they write that the Lpl-Alike protein is *not* able to perform an octanoyl-CoA transfer when measured with another assay. This apparent discrepancy needs to be clarified. Note that if the enzyme has both activities, this would be in line with the presence of two radical SAM enzymes (which might serve in inserting sulfur into the octanoate bound to the LbpA1/A2 proteins. Maybe LplA-like protein is promiscuous and has a dual function? In this respect, some kinetic data on octanoate or lipoate activation would be very helpful to clarify the question of true function or side reactivity. Can the authors estimate rates and/or K_M_ values?

Another, related problem, is that the authors demonstrate that both LbpA1 and A2 are loaded by LplA-like protein. However, they do not answer (or even address!) the question why Thioalkalivibrio features two Lbps (LbpA1 and A2)? Are both of the enzymes important, is the presence of two Lpls functional redundancy or complementarity (e.g. are the two proteins expressed on different substrates, under different conditions or are they forming a heterodimer)? In which respect differ the A1 and A2 proteins from each other? This information would be very useful top the reader who wonders about the presence of two proteins in one cluster. Also, I did not quite understand if the second lipoates ligase outside of the cluster would activate be able to attach the lipoate cofactor to A1 and A2!?

Finally, the authors conclude that "the LplA does not does not recognize lipoyl domains/proteins stemming from other organisms". However, the authors did not show that the lipoate transferase from *Thialkalivibrio* would be able to function with the *H. denitrificans* in my opinion! This statement is not supported/backed up by data.

*Reviewer #2:*

The manuscript describes very interesting follow-up work on the recently discovered heterodisulfide reductase (Hdr)-like multienzyme systems for sulfur oxidation found in a large group of bacteria and archaea: these enzyme systems are resembling HdrABC of methanogenic archaea (but without CoM-CoB heterodisulfide). In another manuscript submitted to ISMEJ, the Ch. Dahl group apparently confirmed the role of these hdr-like genes in oxidation of thiosulfate (as derived from DMS) to sulfate, in their genetically accessible system, *Hyphomicrobium denitrificans*. For the present manuscript, the Dahl lab used the same system, and joined forces with the J.E. Cronan lab, for an exploration of the role of genes for lipoate-binding proteins (LpbA) and lipoate-protein ligases (LplA), which are co-encoded in these hdr-like sulfur-oxidation gene clusters in this organism, in Thioalkalivibrio, and in almost all gene clusters of other organisms with (predicted) Hdr-like sulfur oxidation systems.

First, a comprehensive analysis was done for the potential lipoate-binding protein genes, lipoate-protein ligase genes (and radical-SAM protein genes), which are each co-occurring in such hdr-like gene clusters, essentially in three different arrangements (group 1, 2 and 3). The detailed analysis of the LbpA sequences indicated two strictly conserved Cys residues in comparison to the archetype GcvH of the glycine cleavage system, and two subtypes of LbpAs, i.e. LbpA1 and LpbA2. Then, four different LpbAs were recombinantly produced and tested: both LbpA1 and A2 from a Thioalkalivibrio sp. (Tk-LbpA1 and A2), LbpA1 from a Thiorhodospira sibirica (Ts-LbpA1), and the LbpA2 from H. denitrificans (Hd-LbpA2). Non-lipoylated versus lipoylated proteins obtained dependent on anoxic expression, were distinguished by immunoblotting and by gel-mobility shift. For *H. denitrificans*, specific production of lipoylated HbpA2 during DMS and thiosulfate utilization was demonstrated by proteomics and by immunoassays targeting a representative HdrA2- or the lipoyl-domain. In addition, a HbpA2 in-frame knockout was constructed, which had lost the ability to utilize DMS, and which produced the dead-end product tetrathionate from thiosulfate but not sulfate, thus confirming that HdrA2 is essential for sulfur oxidation (as with the knock-out of the Hdr-like genes).

Next, it was tested whether LpbA1 or A2 can substitute for GcvH in a *Bacillus subtilis* ∆*gcvH* strain (no, they can't) and this failure was attributed to a lack of octanoate-transfer to the LipAs through Bacillus-LipM, as demonstrated in vitro assays using 14C-octanoate as substrate and autoradiography. Further, the ligases Hd-LplA and Tk-LplA cannot substitute for the native ligases of *E. coli* in vivo (in a Δ*lipB* Δ*lplA* knockout strain), and not an in-vitro lipoylation assay using the recombinant proteins. Finally, it was confirmed that the LplAs encoded in the hdr-like gene cluster have activity only with their own LbpAs encoded in the same cluster, by MALDI, autoradiography and mobility-shift assays. This specificity was attributed to a difference in the domain structure of the ligases (absence of auxiliary domain in the LbpAs) and an absence of the octanoyl-transferase activity in these LplAs; it is then speculated whether the LbpAs may confer this function.

Overall, the manuscript provides important, and convincing evidence that lipoate is involved as key component in another metabolic function not previously recognized – sulfur oxidation via the HDR-like system – and that the mechanisms by which lipoate is synthesized and attached to its binding-protein(s) LbpA is more diverse than initially known – here specifically designed to sulfur oxidation. Clearly, one would like to know more on the details of sulfur oxidation route in these Hdr-like enzyme systems, particularly on the electron transfer via the dihydrolipoamide (which might substitute for reverse-electron flow, as postulated in the Discussion), however, this was not the scope of the present study. Rather, the results presented here are (on top of the points mentioned above) also a very important prerequisite for a future characterization of this enzyme system.

I've got no major criticism to raise:

In my opinion, this study is very sound, the results from the two labs allow for all conclusions drawn (as expected), and the topic of the study is clearly in *eLife*'s scope.

---

## [Author Response]

Congratulations on your high quality manuscript, which the Reviewing Editor and two expert reviewers agree should be published in eLife pending some straightforward revision/experimental clarification. The details of the two reviews are appended below. The primary issues we would like you to address are the following:1) Revise the Abstract, Introduction and Discussion so that the main message is clear to the broad eLife audience (recall that many readers may know very little either about lipoate biochemistry or about microbial sulfur metabolic pathways): state specifically what the new function for lipoic acid is that you have discovered. Explain to naive readers why they should care about this. Rather than claiming discovery of a "novel" function, be precise about what the function is. Claims of novelty are overdone and obscure what the exciting finding is.

The title was slightly changed and instead of pointing out a “new role for an old cofactor”, we now mention “an atypical role for an old cofactor”.

The Abstract was rewritten such that it now states the new function of lipoic acid that we discovered: “Here, reverse genetics identified LbpA as an essential component of the Hdr-like sulfur-oxidizing system in the Alphaproteobacterium *Hyphomicrobium denitrificans*. LbpAs likely function as sulfur-binding entities presenting substrate to different catalytic sites of the Hdr-like complex, similar to the substrate-channeling function of lipoate in carbon-metabolizing multienzyme complexes, e.g. pyruvate dehydrogenase.”

In the Abstract we now explicitly state that LbpAs serve a specific function in sulfur oxidation and that they are not modified by the described canonical maturation machineries for lipoyl proteins.

In the Introduction and in the Discussion, we removed the term “novel lipoate-binding proteins” and now refer only to “lipoate-binding proteins”. The likely new function of lipoate within the Hdr-like system of sulfur oxidation is highlighted in the Discussion (second paragraph) and depicted in Figure 8.

2) Clarify the ambiguity pointed out by reviewer 1 regarding octanoyl-transferase activity in two different parts of the text.

This was done. Please refer to our answer to the specific reviewer comment below.

3) Address the specificity of the LplA protein and the role of the A1 and A2 proteins and their redundancy. This could require some additional data.

This was done. Please refer to our answer to the specific reviewer comment below.

Reviewer #1:The manuscript focuses on an microbiologically important and biochemically interesting process: the oxidation of sulfurous compounds in bacteria that proceeds via a protein complex resembling heterodisulfide reductase. This manuscript shows that in Hyphomicrobium denitrificans a lipoylated-protein (LpbA2) is involved. The manuscript also clarifies the question how the lipoate cofactor is installed in LbpA1/A2-proteins (from Thialkalivibrio) through a specific enzyme (LplA-like protein). In general the manuscript contains a lot of high-quality data that supports the chain of evidence. Yet, there are some points that need to be addressed in my opinion.Text/manuscript structure: The authors characterize and investigate proteins from different organisms to draw their conclusions. I wonder whether a slight restructuring of the manuscript to avoid too many "logical jumps" between the systems would be beneficial!? Definitely the authors should add a dedicated table that lists all proteins investigated in this work, as well as their percentage sequence identities to GcvH (for LbpA1/A2) or LplA and/or LipB (for LplA-like protein).

Tables listing all the protein investigated in this work and including their percentage identities to GcvH for LbpA1/A2 and LplA/LipB for the LplA proteins were added (Tables 1 and 2, mentioned in the first paragraphs of the subsections “Production and analysis of LbpA1 and LbpA2 proteins in vivo and in vitro” and “The LplA homologs from sulfur oxidizers cannot replace LplA or LipB in E. coli”.

Data/interpretation: One crucial point is the functional assignment of the LplA-like protein from Thialkalivibrio. If I understand the authors correctly, they claim (subsection “LplA-like protein from sulfur oxidizer bacteria is a bona fide lipoate:protein ligase active on its own lipoate-binding proteins”) that the LplA-like protein is able to perform a lipoylation, as well as octanoylation. A bit later, however, they write that the Lpl-Alike protein is not able to perform an octanoyl-CoA transfer when measured with another assay. This apparent discrepancy needs to be clarified. Note that if the enzyme has both activities, this would be in line with the presence of two radical SAM enzymes (which might serve in inserting sulfur into the octanoate bound to the LbpA1/A2 proteins.

Here we have not been clear enough in the original manuscript. In fact, the Thioalkalivibrio LplA-like protein is able to catalyze an ATP-dependent ligase reaction. In that ligase reaction lipoic acid as well as octanoic acid are accepted as substrates. The reaction consists of two steps. The first is activation of lipoic or octanoic acid by formation of an adenylated form. AMP is hooked up to the substrate, pyrophosphate is released. In the second step, AMP is released and lipoic acid or octanoic acid is hooked up to the acceptor protein.

The octanoyl transferase reaction that was also tested is very different. Here, an octanoyl residue that is already hooked up to a donor protein (either acyl carrier protein or *B. subtilis* GcvH) is simply transferred to an acceptor protein. This reaction is not ATP-dependent. We show, in the subsection “LplA-like protein from sulfur-oxidizing bacteria does not exhibit octanoyltransferase activity”, that the sulfur oxidizer LplA-like proteins do not exhibit this transferase activity. They have this in common with characterized LplA proteins from other organisms that are also enzymes restricted to catalysis of the lipoate:protein ligase reaction. To make these points even clearer to the reader, we introduced a new heading for the aforementioned subsection describing the lack of octanoyltransferase activity for Thioalkalivibrio LplA.

Maybe LplA-lilke protein is promiscuous and has a dual function? In this respect, some kinetic data on octanoate or lipoate activation would be very helpful to clarify the question of true function or side reactivity. Can the authors estimate rates and/or K_M_ values?

As already pointed out above, the LplA-like protein from sulfur oxidizers does not have a dual function. To the best of our knowledge, kinetic data have never been provided for any lipoate:protein ligase or octanoyl transferase. This is due to assay limitations. The assays developed so far and applied in this study allow assessment of substrate consumption and/or product formation by autoradiographic analysis or via gel mobility shifts but are not suited to delineate K_M_ or V_max_ values.

Another, related problem, is that the authors demonstrate that both LbpA1 and A2 are loaded by LplA-like protein. However, they do not answer (or even address!) the question why Thioalkalivibrio features two Lbps (LbpA1 and A2)? Are both of the enzymes important, is the presence of two Lpls functional redundancy or complementarity (e.g. are the two proteins expressed on different substrates, under different conditions or are they forming a heterodimer)?

The reviewer addresses an important point, but unfortunately answers to this question will require a very substantial amount of work that cannot be performed within the framework of this manuscript. We would like to point out that there is no genetic system for any of the organisms featuring two different lbpA genes, so this question cannot be answered by genetic methods. Just as most of the other organisms containing two different lbpA genes,

Thioalkalivibrio is an obligate chemolithoautotroph that is dependent on oxidation of reduced sulfur compounds. We can thus not test, whether the two proteins are expressed on different substrates or under different growth conditions. *H. denitrificans*, the only genetically accessible model organism encoding the Hdr-like system, has only a single lbpA gene and is thus not suited for answering questions on redundancy or complementarity of LbpA1/LbpA2. As clearly stated in the manuscript, sulfur oxidizer LbpA proteins are not lipoylated in an *E. coli* background when produced as recombinant proteins. We therefore refrained from co-production of *Thioalkalivibrio* LbpA1 and LbpA2 because a possible oligomerisation could have been due to artificial interaction of immature apoproteins.

In which respect differ the A1 and A2 proteins from each other? This information would be very useful top the reader who wonders about the presence of two proteins in one cluster.

At the moment it would be premature to delineate different functions for LbpA1 and LbpA2 in those organisms where we have both variants encoded. Sequence comparison does not indicate any conserved differences in functionally important amino acid residues (e.g. cysteine). It should be noted that the well-described α-ketoacid dehydrogenases and acetoin dehydrogenases are enormous protein complexes containing many copies of E1, E2 (the lipoate binding protein) and E3 subunits. The amino-terminal region of each E2 subunit contains one or more small, about 80-aa lipoylation domains. The E2 subunit of pyruvate dehydrogenase contains up to three or these domains. Deletion of one or even two of its three lipoyl domains has no adverse effect on the assembly or catalytic activity nor does the creation of E2 chains containing various combination and permutations of functional and nonfunctional lipoyl domains (Perham 2000). The lipoylation domains are displayed on the outer face of the oxoacid dehydrogenases complexes where they then interact with the E1 and E3 subunits. Acyl group transfer and redox reaction are possible between lipoyl groups on different E2 proteins. A total of three lipoyl domains turned out to be optimal for the E2 subunit of *E. coli* PDH, of which only the outermost needs to be lipoylated, suggesting that the inner two exist principally to extend the reach of the outermost domain and its lipoyl lysine residue (reviewed in Perham 2000). It is likely that LbpAs from sulfur oxidizers serve a similar substrate-channeling function and that more than a single LbpA monomer is present and functional in the mature Hdr-multienzyme system. Gene duplication may have occurred allowing more efficient protein production followed by independent evolution of both genes. The reaction center subunits of phototrophic bacteria are probably the most prominent example for such an evolutionary history. In that case, an original protein homodimer encoded by one gene evolved into a heterodimer of two related yet distinct proteins after gene duplication (Blankenship, 2010). These considerations have been included into the Discussion (third paragraph).

Also, I did not quite understand if the second lipoates ligase outside of the cluster would activate be able to attach the lipoate cofactor to A1 and A2!?

In fact, preliminary experiments were performed with the second lipoate ligase (TK90_0648) from Thioalkalivibrio residing outside of the hdr-like gene cluster. Recombinant TK90_0648 exhibited neither octanoate ligase activity nor octanoate transferase activity upon LbpA1 (TK90_0638) and LbpA2 (TK90_0640). These experiments were performed using hot C^14^ labeled octanoate as the substrate. Given these first negative results we focused on the TK90_0642 encoded ligase which proved to be active with both LbpAs. It is intended to perform further experiments with the second ligase. The outcome of these experiments will be presented in a separate publication in the near future.

Finally, the authors conclude that "the LplA does not does not recognize lipoyl domains/proteins stemming from other organisms". However, the authors did not show that the lipoate transferase from Thialkalivibrio would be able to function with the H. denitrificans in my opinion! This statement is not supported/backed up by data.

In fact, we performed the in vitro assays for LplA-like protein one and LbpA protein from another organism containing a hdr-like gene cluster. This is now mentioned in the second paragraph of the subsection “LplA-like protein from sulfur oxidizing bacteria is a bona fide lipoate-protein ligase active on its own lipoate-binding proteins”, and a gel mobility shift assay showing ATP-dependent octanoylation of Thiorhdodspira LbpA1 catalyzed by LplA from *Hyphomicrobium denitrificans* is now provided as Figure 7—figure supplement 1. The LplA-like protein from *H. denitrificans* showed ligase activity with both of these proteins. Our statement in the Discussion is valid and now supported by the experimental data provided.